



# Diagnosing $O_3$ formation and $O_3$-$NO_X$-VOC sensitivity in a heavily polluted megacity of central China: A multi-method systematic evaluation over the warm seasons from 2019 to 2021

Shijie Yu [a], Hongyu Liu [a], Hui Wang [a]*,Fangcheng Su [b,c], Beibei Wang [a], Minghao Yuan [d],

Kunao Song [a], Zixian Wang [a], Daoqing Xu [a], Ruiqin Zhang [b,c]**

 

 

a. Department of Environmental Engineering, Henan University of Science and Technology, Luoyang 471023, China

b. Institute of Environmental Sciences, Zhengzhou University, Zhengzhou 450001, China

c. School of Ecology and Environment, Zhengzhou University, Zhengzhou 450001, China

d. Environmental Protection Monitoring Center Station of Zhengzhou, Zhengzhou 450007, China

 

 

 

*Correspondence to: H. Wang, Department of Environmental Engineering, Henan University of Science and Technology, Luoyang, Henan, PR China, 471023

**Correspondence to: R. Zhang, Research Institute of Environmental Science, School of Ecology and Environment, Zhengzhou University, Zhengzhou, Henan, PR China, 450001

 

 

E-mail address: wanghui79@haust.edu.cn (H. Wang), rqzhang@zzu.edu.cn (R. Zhang)





**Abstract**

This study investigated the high ozone pollution in Zhengzhou City from 2019 to 2021 using observational data and model simulations, focusing on volatile organic compound (VOC) pollution and its impact on ozone formation. Using online VOC data and statistical analyses, the results showed that VOC concentration increased with ozone pollution level, with average values of 84.7±51.0, 96.6±53.4 and 105.3±59.4 µg/m³ for non-pollution, mildly polluted and moderately polluted periods, respectively. Source apportionment of ozone and its precursor VOCs was performed using CMAQ and PMF models. The results demonstrated that the reduction of vehicle emissions should be prioritized to mitigate ozone pollution in Zhengzhou, since transportation emissions respectively accounted for 64% and 31% of ozone and VOC precursor emissions. In addition, local ozone production rates and HOx base budgets were calculated using an observation-based (OBM) model. The ozone production rates on non-pollution, mildly polluted, and moderately polluted days were respectively 2.0, 4.5, and 6.9 ppbv/h on average. The HOx radical concentration on polluted days was 1.5-6.4 times higher than that on non-pollution days, which is indicative of more efficient radical cycling during photochemical pollution. The $O_3$-NOx-VOC sensitivity was analyzed using the OBM model, CMAQ model and ratio method. The results showed that ozone generation in Zhengzhou was mainly limited by VOCs, suggesting that the reduction of VOCs should be focused on aromatic hydrocarbons and olefins. The optimal reduction ratio of anthropogenic VOCs to NOx was about 2.9:1. This study will offer deeper insights for formulating effective ozone pollution prevention and control strategies.

Keywords: Volatile organic compounds; the Observation-based model; the Community multiscale air quality; the Source apportionment; the $O_3$–$NO_x$–VOC sensitivity.



## 1. Introduction

Recently, China's efforts to combat air pollution have significantly improved air quality across many regions. However, ground-level ozone ($O_3$) concentrations continue to increase, with $O_3$ being the primary pollutant for several days (Chao et al., 2024; Li et al., 2024). Thus, $O_3$ pollution constitutes a pressing challenge. Volatile organic compounds (VOCs) are key precursors of near-surface $O_3$ and are pervasive in the atmosphere. They are precursors of secondary pollutants, such as secondary organic aerosols, peroxyacetyl nitrate, and polycyclic aromatic hydrocarbons, which impact air quality, human health, and vegetation growth (Jia et al., 2024; Kittipornkul et al., 2023; Zhao et al., 2023). Consequently, $O_3$ and its precursors VOCs have attracted significant attention.

Understanding the causes of ozone pollution requires a comprehensive identification of the characteristics and sources of its precursor VOCs. Given the diverse types and concentrations of atmospheric VOCs, the characteristics and variations in VOC concentrations across different regions must be understood for effective VOC management. Researchers have observed distinct spatial and temporal patterns in the concentration and spatiotemporal distribution of VOCs (Mandal et al., 2023; Wang et al., 2022; Zhang et al., 2020). For instance, Huang (2019) collected VOC data from Taichung City for over a year and discovered an average mass concentration of 76.2 μg/m³ (major components = aromatic hydrocarbons (34.8 μg/m³) and alkanes (33.8 μg/m³)). Hui (2018) observed significant daily and seasonal variations in VOC species in Wuhan through the Wuhan VOC integrated observation experiment. The compositions and sources of atmospheric VOCs are complex, varying widely (Xiao et al., 2024; Wu et al., 2024). To manage VOC pollution in urban areas, their sources must be determined. Barletta (2005) identified motor vehicle exhaust and combustion as the primary sources of VOCs via a source analysis in 43 cities across China. Fossil fuel combustion, solvent use, and liquefied petroleum gas (LPG) combustion are predominant VOC sources in the North China Plain. The Yangtze River Delta region is mainly influenced by petrochemical, coal combustion, and rubber industries. Transportation sources significantly contribute to VOC pollution in China (Song et al., 2018; Yu et al., 2022). VOC management requires targeted strategies to control such sources. Research indicates that VOC pollution characteristics and source compositions markedly vary across cities, posing challenges to VOC control and



photochemical pollution management. Although comprehensive VOC databases exist
for economically developed regions and major urban agglomerations such as
Beijing–Tianjin–Hebei, the Yangtze River Delta, and the Pearl River Delta, detailed
research on central provinces, such as Henan, is limited, particularly studies based on
long-term observations. This limits the efficiency of local VOC prevention and
control and hampers local photochemical pollution control. Thus, in-depth studies on
the VOC pollution characteristics in such under-researched regions are required to
improve local air quality and reduce photochemical pollution.
With the escalating $O_3$ pollution, understanding the mechanisms behind $O_3$ generation
is important in mitigating local photochemical pollution (Liao et al., 2024; Liu et al.,
2021). $O_3$ generation is generally categorized into nitrogen oxides ($NO_x$) control,
VOC control, and transition zones (Sillman, 2021). From a photochemical perspective,
the division into $O_3$–$NO_x$–VOC control zones is largely determined by the relative
contributions of $NO_x$ and OH sources. Urban areas, such as New York (Tran et al.,
2023), London (Tudor, 2022), Mexico City (Santiago et al., 2024), Beijing (He et al.,
2022), Shanghai (Liu et al., 2021), Guangzhou (Hong et al., 2022), and Chengdu (Tan
et al., 2018), are predominantly in VOC control zones. Contrarily, remote rural areas,
such as Okinawa, Japan (Martin et al., 2004), Taian in Shandong Province (Li et al.,
2024), and Wangdu in Hebei Province (Ran et al., 2011), are more in $NO_x$ control
zones. Beyond these spatial differences, $O_3$ generation sensitivity exhibits significant
temporal variation. Liu (2010) observed that in January, most areas in eastern China
were in VOC control zones, whereas by July, they shifted to $NO_x$ control zones. Pan
(2015) observed that daily variations showed VOC control in the morning and a
transition to $NO_x$ control in the afternoon. This highlights the complexity of
near-surface $O_3$ formation and the highly nonlinear relationship between $O_3$ and its
precursors, constituting a major challenge in $O_3$ pollution control. Owing to the spatial
and temporal variability in $O_3$–$NO_x$–VOC sensitivities, the primary factors
influencing local $O_3$ generation must be investigated. Given the various methods for
studying $O_3$–$NO_x$–VOC sensitivity, with their specific strengths and limitations, a
comprehensive approach using multiple methods is essential to better understand local
$O_3$ production.
In Zhengzhou, Henan Province, with a population above 13 million in 2023 and
ranking sixth in the nation with 5.0 million vehicles, the city experiences severe haze
and photochemical pollution due to air pollutant emissions (Wang et al., 2021; Zhang



et al., 2023). Although significant progress has been made in controlling primary
pollutants in the Central Plain Urban Agglomeration, with Zhengzhou at its core,
evidenced by meeting the national standards for $SO_2$, $NO_2$, and CO, and a decrease in
the annual average of $PM_{2.5}$, the $O_3$ concentration in Zhengzhou continues to increase.
Thus, photochemical pollution must be urgently addressed (Min et al., 2022; Yu et al.,
2021). $O_3$, a potent oxidant, impacts air quality and compromises human immune
function and health. For effective prevention and control, the causes of $O_3$ pollution in
Zhengzhou must be analyzed. This can be achieved by characterizing the sources and
characteristics of ozone and its precursors, understanding the local ozone formation
mechanisms and proposing appropriate reduction strategies. However, local studies
remain limited, focusing on single-site and short-term VOC and $O_3$ concentrations
(Jia et al., 2024; Li et al., 2020), with only a few addressing critical factors
influencing $O_3$ generation. Therefore, it is very necessary to investigate the
characteristics of volatile organic compounds and effects on ozone pollution.
Here, we used online VOC monitoring data spanning 2019 to 2021, integrating the
Ozone Box Model (OBM) and Community Multiscale Air Quality (CMAQ) model to
investigate the $O_3$ photochemical generation mechanisms in Zhengzhou. The
objectives were (1) to analyze the pollution characteristics of atmospheric VOCs in
Zhengzhou and clarify the temporal distribution differences, (2) identify the $O_3$ and
VOC sources, and (3) investigate localized $O_3$ photochemical generation and removal
pathways. Finally, we established the $O_3$ isoconcentration curves of Zhengzhou;
elucidated the relationship among $O_3$ generation, VOCs, and $NO_X$; and proposed
targeted control measures for $O_3$ pollution.
**2. Observation and methodology**
**2.1 Monitoring stations and instruments**
The municipal environmental monitoring station (MEM; 113°36′E, 34°45′N) was
selected as the study site to obtain real-time online data, covering May to September
during the period from 2019 to 2021 (Fig. S1). Located on the roof of a four-story
building at the municipal environmental monitoring station, the area of the sampling
site is predominantly commercial and residential, with no significant industrial
sources nearby. The station is situated 300 m west of Qinling Road and 200 m south
of Zhongyuan Road, both of which experience heavy traffic. Thus, mobile sources



may significantly contribute to the VOC concentration of the site. The MEM station is
part of the air monitoring network operated by the Zhengzhou Environmental
Monitoring Center. Simultaneously observed meteorological parameters include
temperature, relative humidity, atmospheric pressure, wind direction, wind speed, and
trace gases, such as $O_3$, NO, and $NO_X$.
Here, VOC data with a temporal resolution of 1 h were collected using a Wuhan
Tianhong online monitoring system (TH-300B), comprising two main modules (a
cryogenic preconcentration system and gas chromatography/mass spectrometry
(GC/MS) system). The cryogenic preconcentration unit employs an electronic
refrigeration technique, achieving an extreme internal temperature of −150°C in the
cold trap, effectively capturing the target compounds. This low-temperature empty
tube trapping method is advantageous over traditional techniques because it
minimizes the disadvantages of adsorbent adsorption, reduces VOC loss, and
enhances data accuracy. Teflon tubes were used to prevent chemical interference from
adsorbents. Prior to air sample collection, a water removal device was used to
eliminate excess moisture, preventing VOC loss during low-temperature
preconcentration. A particulate removal device was installed at the inlet of the
sampling tube to filter out airborne particulate matter.
The complete workflow of the monitoring system includes sample collection, freeze
trapping, thermal desorption, GC-flame ionization detector (FID)/MS analysis, and
heating. To ensure the accuracy of the data obtained during the observation, rigorous
quality assurance and quality control measures were implemented (Wang et al., 2022).
Prior to analytical testing, the GC-FID/MS system was periodically calibrated using
an external standard gas across five concentration gradients (0.8, 1.6, 2.4, 4.0, and 8.0
ppbv), generating five-point calibration curves for each analyte. Four internal standard
gases   (bromochloromethane,   1,4-dichlorobenzene,   chlorobenzene,   and
fluorobromobenzene) were used to ensure instrument stability. The linear correlation
coefficients ($R^2$) for the VOCs measured using the instrument exceeded 0.99, and the
method detection limits (MDLs) were in the range of 0.003–0.121 ppbv (Yu et al.,
2021; Wang et al., 2022). Notably, 90% of the target compounds exhibited a
quantification accuracy within 25%, and the measurement precision, as indicated by
the relative standard deviation of the peak area, was maintained below 5%.



**2.2 Analytical model for O₃ formation mechanism**

**2.2.1 Relative incremental reactivity**

Here, we employed the OBM, commonly used in transformation studies of atmospheric VOCs to investigate the formation of $O_3$, free radicals, and intermediates (Niu et al., 2024; Zhou et al., 2024). The atmospheric chemistry framework employed here was based on the Master Chemical Mechanism (MCM) v3.3.1 (http://mcm.leeds.ac.uk/mcm/), which describes the degradation processes of methane and 142 non-methane VOCs, encompassing over 17000 reactions and 5800 substances and radicals (Chen et al., 2023; Fu et al., 2024).

The model inputs include 61 VOCs and 8 oxygenated VOCs (including acrolein, acetone, 2-butanone, 4-methyl-2-pentanone, 2-hexanone, 2-propanol, ethyl acetate, methyl tert-butyl ether (MTBE), and 1,4-dioxane), along with inorganic trace gases ($NO_x$, $SO_2$, and CO) and relevant meteorological factors (temperature, barometric pressure, and relative humidity) to constrain the model. Owing to the unavailability of measured photolysis rate parameters (j values), we simulated these parameters using the Tropospheric Ultraviolet and Visible (TUV) model (TUVv5.2, http://cprm.acom.ucar.edu/models/TUV), which is widely employed for such applications. The simulation period was set to 05:00–19:00 within the observation time frame.

The relative importance ratios (RIRs) were computed using the OBM to assess the relationship between $O_3$ precursors (Chai et al., 2023; Hu et al., 2023), as follows:

$$RIR^S(X) = \frac{\left[P_{O_3}^S(X) - P_{O_3}^S(X - \Delta X)\right]/P_{O_3}^S(X)}{\Delta S(X)/S(X)} \tag{1}$$

where X is a specific precursor, S(X) is the actual concentration of substance X, and $\Delta S(X)$ is the theoretical change in S(X). $P_{O_3}^S(X)$ and $P_{O_3}^S(X-\Delta X)$ refer to the simulated $O_3$ yields based on varying the concentration of species X in baseline and theoretical scenarios, respectively.

Net $O_3$ production was simulated based on the OBM model. The net $O_3$ production rate ($P_{O_3}^S$) is the difference between the gross $O_3$ production rate ($G_{O_3}^S$) and destruction rate ($D_{O_3}^S$).

$$P_{O_3}^S = G_{O_3}^S - D_{O_3}^S. \tag{2}$$



$G_{O_3}^S$ was calculated by accumulating the oxidation rates of NO by $HO_2$ and $RO_2$.
$$G_{O_3}^S = k_{HO_2+NO}[HO_2][NO] + \sum k_{RO_{2i}+NO}[RO_{2i}][NO] \tag{3}$$

In addition, $D_{O_3}^S$ was calculated based on $O_3$ photolysis, reactions between $HO_2$ and
olefins, and reactions between $O_3$ and OH and between $NO_2$ and OH.
$$D_{O_3}^S = k_{O(^1D)+H_2O}\left[O(^1D)\right][H_2O] + k_{OH+NO_2}[OH][NO_2] + k_{OH+O_3}[OH][O_3] +$$
$$k_{HO_2+O_3}[HO_2][O_3] + k_{alkenes+O_3}[alkenes][O_3] \tag{4}$$

The values of the intermediates and radicals were obtained from the output of the
OBM model. Constants k in Eqs. (3) and (4) are the rate coefficients for the matching
reactions, respectively.

### 2.2.2 Empirical kinetic modeling approach

The empirical kinetic modeling approach (EKMA) was developed based on OBM
calculations, commonly employed to assess the sensitivities of $O_3$ to $NO_x$ and VOCs
(Liang et al., 2023; Liu et al., 2022). This approach was employed to characterize the
nonlinear relationship between $O_3$ and its precursors. Considering that the mixing
ratio of VOCs does not accurately reflect the amount of $O_3$ produced, the VOC
concentration was substituted with the total OH reactivity of anthropogenic
hydrocarbons in generating the EKMA curves. The reactivity was calculated using the
hydroxyl radical (OH) reaction constants corresponding to the concentrations of
VOCs and $NO_x$ in the model. By varying the concentrations and reactivities of the
VOCs and $NO_x$, the precursors were identified as a function of the $O_3$ production rate
$P_{O_3}^S$, leading to the generation of the EKMA curves. The net $O_3$ production rate $P_{O_3}^S$,
$O_3$ generation rate $G_{O_3}^S$, and $O_3$ depletion rate $D_{O_3}^S$ were calculated using Eqs. (2)–(4),
respectively.

### 2.2.3 Decoupled direct method

The decoupled direct method (DDM) can be employed to analyze the sensitivity of $O_3$
to its precursors. By directly solving the sensitivity equations of the air quality model,
various sensitivity coefficients can be obtained, enabling comprehensive sensitivity
analyses. The primary objective of this study was to calculate the semi-normalized
first- and second-order sensitivities of $O_3$ concentration with respect to anthropogenic




VOCs and NO$_x$ emissions.

$$S_V = \frac{\partial C_{O_3}}{\partial V}. \tag{5}$$

$$S_N = \frac{\partial C_{O_3}}{\partial N}. \tag{6}$$

$$S_{VV} = \frac{\partial^2 C_{O_3}}{\partial V^2}. \tag{7}$$

$$S_{NN} = \frac{\partial^2 C_{O_3}}{\partial N^2}. \tag{8}$$

$$S_{VN} = \frac{\partial^2 C_{O_3}}{\partial V \partial N}. \tag{9}$$

In the model, V, N, and O$_3$ represent VOCs, NO$_x$, and O$_3$, respectively. C$_{O3}$ is the O$_3$
concentration, whereas $\varepsilon_V$ and $\varepsilon_N$ are the relative perturbations in total anthropogenic
VOCs and NO$_x$ emissions from sources in Henan Province. The first-order
sensitivities of the O$_3$ concentration to NO$_x$ and VOC emissions are denoted by S$_N$
and S$_V$, respectively. S$_{NN}$ and S$_{VV}$ denote the second-order sensitivities.
For model calibration using CMAQ-DDM, the pollutants considered for validation
included routinely observed meteorological parameters and pollutant concentration
data. Table S1 shows that the simulated PM$_{2.5}$ concentrations were slightly
overestimated, with an overall overestimation of approximately 20%, closely aligning
with the actual air quality conditions. The simulation results for O$_3$ revealed a better
performance than those for PM$_{2.5}$, achieving an overall correlation (R) of 0.74, which
met the requirements for targeted research on O$_3$ during the study period. However,
the O$_3$ concentrations were slightly underestimated, with an overall underestimation
of 17%. Conversely, the NO$_2$ simulation was generally more accurate, indicating a
strong sensitivity to NO$_x$ and VOC emissions. This consistent underestimation of O$_3$
suggests that VOC emission sources may be underestimated to an extent, although
NO$_2$ simulations are largely accurate.
Overall, the simulation results effectively reproduced the spatial and temporal
distribution characteristics of air pollution in Henan Province and Zhengzhou City,





providing a solid foundation for research (Su et al., 2021).
**2.3 Pollutant source attribution**
**2.3.1 O$_3$ source apportionment**
Traceability analysis is employed to uniquely label or add tracer ions to single
emission substances from different regions or industries within the emission source
inventory (Xian et al., 2024; Zhang et al., 2023). Thus, our CMAQ (Source Oriented
CMAQ v5.3.2) model, which includes a source apportionment function,
independently computed the scientific processes affecting pollutants marked with
unique industry or regional labels (Su et al., 2023). By analyzing the concentration
results of these labeled species, we determined the contribution of each pollutant.
In the regional traceability simulation, we quantified the contributions of various
emission sources in Zhengzhou through industry traceability (as shown in Fig. S2).
Specifically, we simplified the emission sources (originally categorized into 16 types)
into seven categories based on a refined 1-km inventory of Zhengzhou City. Thus, the
contributions of local emissions to O$_3$ formation from different industries were
determined, facilitating more accurate industry and regional control.
**2.3.2 VOC source apportionment**
The Positive Matrix Factorization (PMF) model (version 5.0) developed by the U.S.
Environmental Protection Agency was used for the source apportionment of VOCs
(Farhat et al., 2024; Frischmon et al., 2024). PMF is a multivariate factor analysis tool
that decomposes measurement data into source profile and source contribution
matrices. As shown in Eq. (10), the quality of species can be determined using the
contribution of the source to the target source and the species distribution of each
source:
$$X_{ij} = \sum_{k=1}^{p} g_{ik} f_{kj} + e_{ij} . \tag{10}$$

X$_{ij}$ is the concentration of the jth substance measured in the ith sample, $g_{ik}$ is the
contribution of the kth source to the ith sample, $f_{kj}$ is the proportion of the jth
substance in the kth source, and $e_{ij}$ is the residual amount of the jth substance in the
jth sample.
The results obtained using Eq. 10 typically present uncertainty (UNC) because of the
error fraction of the species concentration and MDL. PMF analysis relies on the



objective function (Q) to minimize residuals and uncertainties.

$$Q = \sum_{i=1}^{n} \sum_{j=1}^{m} \left[ \frac{x_{ij} - \sum_{k=1}^{p} g_{ik} f_{kj}}{u_{ij}} \right]^2 \qquad (11)$$

where n and m are the numbers of species and samples, respectively, and $u_{ij}$ is the
UNC of the jth species in the i-th sample. Q (true) is the fitting parameter calculated
when all the data are included, whereas Q (robust) is calculated when the model
excludes inappropriate data. Q can be used to select the best mathematical result.
The calculation method for UNC related to the PMF model is as follows:
If the sample concentration is less than or equal to the MDL, the UNC is calculated
using Eq. (12):

$$unc = \frac{5}{6} \times MDL . \qquad (12)$$

If the concentration exceeds the MDL, UNC is calculated using Eq. (13):

$$unc = \sqrt{\left( ErrorFration \times Conc. \right)^2 + \left( 0.5 \times MDL \right)^2} \qquad (13)$$

where unc refers to the UNC of species in the sample, MDL refers to the minimum
limit of the detection method, and Error refers to the sample error (typically
10%–20%, set to 10% here).
The species selection of PMF is based on (1) selecting VOC species that can indicate
specific pollution sources (e.g., in the case of isoprene). (2) The selection of such
VOC species is based on their high environmental concentration levels, source
indication, and signal-to-noise (S/N) values above 5 (i.e., S/N > 5). Thus, prior to
source analysis, the data were screened and processed as follows: (1) VOC species
with lower concentrations were excluded because of the high frequency of
concentrations below the MDL of the instrument (Zhou et al., 2019). (2) Owing to
local emissions, abnormally high concentrations can significantly impact the final
results. Thus, certain species with abnormally high concentrations were excluded.
The final selection comprised 37 VOC specie. The total concentrations of these VOCs
accounted for 82% of the overall VOC concentration, indicating that they effectively
represented the main VOC situation. To determine the number of factors for PMF
resolution, 4–10 factors were tested. The change in Q(robust) indicated a significant
decrease with an increase in the number of factors from 5 to 6. However, the decrease
from 6 to 7 was minimal. When the number of factors exceeded 6, one of the resolved



factors was decomposed into multiple factors that could not be attributed to a single
source. Consequently, the final number of factors was established at 6.

### 3. Results and discussion

### 3.1 General characteristics

### 3.1.1 Levels of air pollutants and meteorological parameters

The results indicate severe photochemical pollution in the period from May to
September during the years 2019-2021 (in Fig. 1), with frequently high $O_3$ values.
The proportion of the maximum daily average of 8-h $O_3$ concentration (MDA8)
exceeding the national secondary standard limit was as high as 45%, with moderate or
higher pollution days accounting for 7%. The MDA8 was recorded on June 6, 2021
(285 μg/m³), with a severe pollution level. During this period, particulate matter
pollution was relatively light, with $PM_{2.5}$ exceeding the national secondary standard
limit only on 2 days ($PM_{2.5}$ daily average concentration > 75 μg/m³). The proportion
of MDA8 exceeding the standard during high $O_3$ pollution periods from 2019 to 2021
was 53%, 37%, and 36%, showing a downward trend but still indicating severe $O_3$
pollution.
$O_3$ concentration is significantly influenced by meteorological factors. Fig. S3 lists the
pearson correlation coefficients between $O_3$ and meteorological factors. $O_3$ was
positively correlated with temperature and wind speed, with correlation coefficients of
0.66 and 0.43 (p < 0.01), respectively. It was negatively correlated with relative
humidity (p < 0.01), with a correlation coefficient of −0.23.
The generation of ozone is intricately tied to the emissions of its precursor gases. As
illustrated in Fig. S3, $O_3$ demonstrates notable correlations with its precursors, namely
VOCs, NO and $NO_2$, exhibiting correlation coefficients (r) of -0.28, -0.30, and -0.57,
respectively. Conversely, $PM_{2.5}$ displays significant positive correlations with these
same three precursors, with r values of 0.36, 0.17, and 0.38, respectively. The positive
relationship between $PM_{2.5}$ and its precursor emissions suggests that regulating these
emissions can effectively mitigate the concentration of particulate matter (Shao et al.,
2024). However, the formation process of ozone is markedly complex, and the



reduction of precursor emissions might not invariably alleviate photochemical
pollution, thereby posing a more formidable governance challenge (Wang et al., 2024).
Consequently, the subsequent sections of this paper will delve into a comprehensive
analysis of the formation mechanism and sensitivity of $O_3$-NOx-VOC.

**3.1.2 $O_3$ vs. non-$O_3$ episode days**

Meteorological conditions are a significant factor in $O_3$ pollution. High temperatures,
low relative humidity, and weak winds facilitate photochemical pollution (Xu et al.,
2023). Nighttime short-term rainfall does not necessarily alleviate photochemical
pollution the next day. Thus, there may be rainfall at night but moderate $O_3$ pollution
the next day (Du et al., 2024). Apart from meteorological factors, $O_3$ precursors
significantly affect $O_3$ formation. Table 1 shows that $NO_x$ concentrations during
pollution periods significantly exceeded those during non-pollution periods, and the
concentration increased as pollution intensified. $PM_{10}$ exhibited a similar pattern,
suggesting that during photochemical pollution periods, the emission intensity of
atmospheric pollutants is relatively high (Saqer et al., 2024). The daily average
concentration of $PM_{2.5}$ on polluted days significantly exceeds that on non-pollution
days. During the sampling period, the average $PM_{2.5}$ concentration on moderate
pollution days was $35.5 \pm 16.4$ µg/m³, closed to the annual average ambient air quality
standard (GB 3095-2012) Grade II standard of 35 µg/m³ and 1.3 times higher than
that during non-pollution periods. NO concentration decreased with increasing
pollution severity. On moderately polluted days, the average NO was 1.3 µg/m³,
which was lower than 7% of the levels observed on non-polluted days. This is
because, on $O_3$-polluted days, the atmospheric oxidation capacity is strong, and $O_3$
and free radical concentrations are high, rapidly consuming NO (Shao et al., 2024).
The average VOC concentrations for non-pollution, light pollution, and moderate
pollution periods were $84.7 \pm 51.0$, $96.6 \pm 53.4$ and $105.3 \pm 59.4$ µg/m³, respectively.
Considering the numerous VOC types and sources, the top 20 substances for the three
stages were analyzed. Table 2 and Fig. S4 show that during the observation period,
higher concentrations of small-molecule hydrocarbons, such as ethane and propane,



suggest a significant influence of LPG/natural gas (NG) at the monitoring site
(Derwent et al., 2017). The acetylene and 1,2-dichloroethane concentrations increased
as pollution intensified, indicating a substantial impact from combustion, particularly
during photochemical pollution (Zuo et al., 2024). The concentrations of C4–C5
alkanes and benzene series compounds were high, suggesting an association with
vehicle emissions (Han et al., 2024). Furthermore, on moderate pollution days,
vehicle tracer substances were more concentrated. The concentrations of n-hexane,
dichloromethane, trichloromethane, tetrachloroethylene, and ethyl acetate were high,
indicating emissions from solvent use. During pollution periods, the isoprene,
2-butanone, and 2-hexanone concentrations exceeded that during non-pollution
periods, indicating a significant impact of plants and more photochemical secondary
products during high $O_3$ periods.

### 401    3.1.3 Diurnal variation

Fig. S5 illustrates the $O_3$ diurnal cycle divided into the suppression phase (P1) of $O_3$
by midnight and early morning NO emissions, photochemical generation phase (P2)
of $O_3$, and titration phase (P3) of $O_3$ by precursor substances prior to the evening peak
(Du et al., 2024). To reflect the photochemical processes, we examined the ratios of
compounds with different reaction rates of $K_{OH}$ radicals but similar sources. Fig. S5
shows the relationship between ethylbenzene and xylene, revealing their homogeneity
($R^2 > 0.9$). Ethylbenzene has a lifetime of 3 days, whereas the lifetime of xylene is 1
day. During the observation period, the diurnal variation trends of $O_3$ and the
ethylbenzene/xylene ratio were similar. The strong correlation between the age
indicator (the ratio of both compounds) and $O_3$ provides strong evidence linking $O_3$
with the photochemical processes of non-methane hydrocarbons (Hui et al., 2019).
The mean $O_3$ concentrations on non-polluted, lightly polluted, and moderately
polluted midnights are $82.9\pm50.3$, $121.4\pm78.2$, and $149.4\pm80.7$ µg/m³, respectively.
As shown in Fig. S5, on polluted days, high fresh NO emissions at midnight
significantly reduced the $O_3$ levels. The concentration decreased to a minimum in the
early morning because of the high fresh NO emissions during the morning rush hour.





Contrarily, on non-polluted days, the NO concentration was lower at night, leading to
a weaker titration effect. Thus, the $O_3$ concentration remained relatively stable at
midnight and decreased to its minimum with the morning peak. Fig. 2 demonstrate the
daily variation trends of characteristic VOCs. In stage P1, the concentrations of NO,
$NO_2$, and n-pentane increase on polluted days, whereas these pollutants remain
relatively stable on non-polluted days. Furthermore, benzene series compounds
(toluene, ethylbenzene, and meta/para-xylene) exhibit similar patterns. Thus, we can
infer that on polluted days, nighttime emissions are significantly influenced by motor
vehicle emissions (Song et al., 2023).
Stage P2 is the accumulation phase of $O_3$ (08:00–16:00). Photoreactions generate
many peroxy radicals (such as $HO_2$ and $RO_2$), converting sufficient NO into $NO_2$
(Fittschen, 2019). With increasing solar radiation, a large amount of $NO_2$ is
photolyzed to generate $O_3$, causing peak $O_3$ levels in this phase. The increase in $O_3$
during polluted periods significantly exceeded that during non-polluted periods. For
example, at 08:00 in non-polluted periods, the $O_3$ concentration was 47.2 µg/m³,
increasing to 59.3 µg/m³ the next moment. During lightly and moderately polluted
periods, the $O_3$ concentration ranges were 56.4–81.5 and 70.8–104.2 µg/m³,
respectively. This is because higher NO concentrations at night during polluted
periods contributed to the formation of more OH and peroxy radicals. Furthermore,
$O_3$ began to accumulate with increasing light intensity. Owing to the higher radical
content, the photochemical reaction activity was strong, and the high concentration of
peroxy radicals further oxidized NO to $NO_2$, leading to higher $O_3$ generation
efficiency during polluted periods. The minimum value of $O_3$ during moderately
polluted periods was observed at 07:00, and although the NO concentration remained
high (13.6 µg/m³), the $O_3$ concentration had already accumulated and rapidly
increased at 09:00. This indicated that the radical concentration was high, rendering it
advantageous in competing with NO for $O_3$, leading to an increase in the $O_3$
concentration. As the NO titration weakened and photochemical activity increased, $O_3$
rapidly accumulated. Throughout the P2 stage, the concentrations of $O_3$ precursors
decreased because of the consumption in photochemical reactions. As shown in Fig. 2,



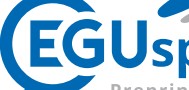

the concentrations of certain VOCs (particularly benzene series compounds) and NO
were lower on polluted days than on non-polluted days in this stage. It should be
noted that isoprene peaks around noon, owing to temperature- and light-dependent
emission rates. For the sensitivity of daily variation patterns, the ratio gradually
increased in P2, suggesting that the increase in the surrounding biogenic VOCs
(BVOCs) may shift the $O_3$ generation mechanism from VOCs to a coordinated control
zone at noon (Chen et al., 2022).
With the arrival of the evening peak and a gradual decrease in light intensity, the $O_3$
sources were less than the sinks, resulting in a continuous decrease in $O_3$
concentration in P3. The NO concentration during P3 was significantly lower than
that during P1. Owing to the effective NO titration, the $O_3$ concentration during P3 on
polluted days exceeded that on non-polluted days. For example, at 20:00 on a
moderately polluted day, the average $O_3$ concentration remained high at 176.3 µg/m³,
decreasing to 131.5 µg/m³ by 23:00. This resulted in stronger atmospheric oxidation
at midnight and higher radical reaction activity (An et al., 2024). This influenced the
$O_3$ generation the next day, contributing to consecutive $O_3$ pollution days.
**3.2 Source apportionment of $O_3$ and VOCs**
**3.2.1 Source contributions to $O_3$**
Fig. 3 shows the spatial distribution of anthropogenic $O_3$ emissions in Zhengzhou. As
shown in Fig. 3a, local anthropogenic emissions contribute significantly to $O_3$ levels
in Zhengzhou, with the overall MDA8-$O_3$ (8-hour maximum daily average ozone
concentration) exceeding 28 ppbv (55 µg/m³), and concentrations exceeding 40 ppbv
observed in urban areas. Fig. 3b shows the anthropogenic contributions in other
regions of Henan Province, revealing that areas surrounding Zhengzhou also have
relatively high $O_3$ concentrations, reflecting the regional nature of ozone pollution.
This phenomenon suggests that controlling ozone pollution is challenging and
requires cross-regional mitigation measures.
To identify the main sources of the significant increase in MDA8-$O_3$, we categorized
the total contributions into six sectors: industries, solvents, transportation, electricity,



residential areas, and other sources (Fig. 3c–h). The results indicate that transportation
is the largest contributor to O$_3$ formation in Zhengzhou, with the highest
concentrations exceeding 30 ppbv in the eastern part of the city. This area is
characterized by a dense road network, including several national expressways, which
suggests a close relationship between transportation emissions and high ozone levels
(Gu et al., 2019). Industrial emissions are the second largest source, with high ozone
concentrations primarily found in the northern and northwestern parts of Zhengzhou,
aligning with the city's industrial layout. The power sector also contributes to ozone
formation in Zhengzhou, with a concentration peak observed in the southwestern part
of the city.
Fig. S6 shows the contributions of various sources to O$_3$ formation during both
polluted and non-polluted periods of the observation. Transportation accounted for
63.6% of total contributions, but this decreased to 57.4% on polluted days. Traffic
sources are the largest contributors to O$_3$, consistent with previous research findings
(Cheng et al., 2019; Su et al., 2023). Industrial sources contributed 30.4% on average
and 26.4% on polluted days, indicating a slight reduction during pollution events.
Electricity contribution significantly increased on polluted days, reaching 3.3 times
the average. Simulation results suggest that more aggressive control measures are
required in the transportation and industrial sectors during the summer. Attention
should also be given to the power sector due to its increased emissions on polluted
days, in order to mitigate O$_3$ pollution under unfavorable conditions.
**3.2.2. Source apportionment of ambient VOCs**
Fig. 4 shows the source apportionment factor spectrum for high O$_3$ pollution periods.
Six factors were identified.
Factor 1 is characterized by dominant species, including acetylene (63%),
chloromethane (25%), benzene (15%), and certain lower-carbon hydrocarbons
(isobutane, n-pentane, ethylene, propylene, and trans-2-pentene, etc.). Acetylene,
ethylene, and chloromethane are important indicators of fossil fuel and biomass
combustion (Liu et al., 2008; Wu et al., 2016). Fixed combustion sources are major



sources of C2–C3 lower-carbon alkanes and benzene (Li et al., 2024); thus, Factor 1 is identified as a combustion source.

Factor 2 mainly comprises C2–C5 alkanes, including ethane (56%), propane (48%), n-butane (37%), isobutane (31%), n-pentane (15%), and isopentane (19%). These substances are tracers for fuel evaporation (gasoline and LPG/NG) (Zhang et al., 2019). Pentane is one of the most abundant VOC species associated with gasoline evaporation (Zhang et al., 2019)., and butane has been reported as a tracer for LPG (Liu et al., 2008; Shen et al., 2018). Furthermore, the aromatic content in this source is extremely low; thus, this source is identified as LPG/NG.

Factor 3 is characterized by high MTBE levels (54%), small-molecule hydrocarbons (C2–C6), and benzene series compounds. C2–C6 alkanes, alkenes, and benzene series compounds are typical tracers of motor vehicle exhaust (Xiao et al., 2023; Wu et al., 2023). MTBE is commonly used as an additive in gasoline, which improves the octane rating, enhances engine performance, and reduces exhaust emissions, making it a tracer for motor vehicle exhaust (Schifter et al., 2017). Thus, Factor 3 is determined to be a motor vehicle emission source.

Factor 4 is characterized by high C6–C8 alkane levels, such as n-hexane (68%), 3-methylpentane (29%), 2-methylpentane (30%), and n-heptane (26%). This factor features high levels of acetone (59%), dichloromethane (45%), chloroform (31%), carbon tetrachloride (69%), and benzene series compounds. Studies have shown that benzene series compounds commonly originate from solvent-use emissions (Zhou et al. 2019, Wang et al. 2021). Carbon tetrachloride, n-hexane, and dichloromethane are commonly used chemical reagents. The content of highly volatile small-molecule hydrocarbons in Factor 4 is low. Thus, Factor 4 is identified as a solvent source.

Factor 5 is characterized by pollutants, mainly comprising small-molecule hydrocarbons (propane, butane, ethylene, and propylene, etc.), BTEX (VOC group including benzene, toluene, ethylbenzene, and xylene), carbon disulfide, and halogenated hydrocarbons (carbon tetrachloride, dichloromethane, chloromethane, and 1,2-dichloroethane, etc.). These substances are widely used in manufacturing, furniture, shoe, and rubber industries (Hui et al., 2018; Yu et al., 2021); thus, Factor 5



is identified as an industrial source.
Factor 6 has the highest proportion of isoprene (89%), which is a marker of plant
emissions ( Cheun et al., 2014; Khruengsai et al., 2024;); thus, Factor 6 is identified as
a plant source.
Fig. 5 shows the source apportionment of VOCs at different $O_3$ pollution levels from
June to September. The results indicate that the motor vehicle emissions gradually
increased as $O_3$ pollution intensified. On moderately polluted days, this source
accounted for up to 35%; thus, during high $O_3$ pollution periods, it is crucial to
enhance the control of motor vehicle emissions. The proportion of combustion
sources is significantly higher on moderately polluted days compared with other
periods. Therefore, combustion sources must be controlled during $O_3$ pollution
periods. On moderately polluted days, the proportion of plant sources is relatively
high, closely related to high temperatures and strong radiation. During the observation
period, the proportion of solvent-use sources was high but decreased with increasing
pollution. It is speculated that solvent emissions, which include highly active aromatic
hydrocarbons, are consumed more because of the high photochemical activity during
$O_3$ pollution periods. Summarily, during high $O_3$ pollution periods, attention should be
focused on controlling motor vehicle, solvent use, and combustion sources.
**3.3 Differences in photochemical reactivity**
**3.3.1 In situ net $O_3$ production**
As shown in Fig. 6, the mean of net $P(O_3)$ during daytime (05:00–19:00) in
Zhengzhou City during the observation period was 3.1 ppbv/h. This was lower than
that of Beijing (5.8 ppbv/h) (Xue et al., 2014), Wuhan (6.2 ppbv/h) (Lu et al., 2017),
and Taishan (4.2 ± 0.9 ppbv/h) (Kanaya et al., 2009), and higher than that of Shanghai
(Liu et al., 2021) (2.8 ± 0.7 ppbv/h), etc. Fig. 6 shows the daily $P(O_3)$ trends for
different pollution periods. The average net $P(O_3)$ during the daytime averages were
2.0 (non-polluted), 4.5 (mildly polluted), and 6.9 ppbv/h (moderately and highly
polluted), which were converted into the $O_3$ daytime chemical accumulation, with
values of 30, 67, and 103 ppbv, respectively. To elucidate the local $O_3$ photochemical



generation and removal pathways, the source–sink pathways and their corresponding
shares in the $O_3$ generation process were investigated.
As shown in Table 3, the mean $F(O_3)$ in Zhengzhou was 3.8 ppbv/h, mainly controlled
by three pathways, with contributions of 84% ($RO_2+NO$), 16% ($HO_2+NO$), and <1%
($MO_2+NO$), respectively. The mean $O_3$ removal rate $D(O_3)$ was 0.7 ppbv/h; the
contribution of $OH+NO_2$ and $O_3$+alkenes contributed the most to the $D(O_3)$ with 56%
and 33%, respectively. The distribution of $O_3$ generation and removal pathways in
different periods and years were statistically analyzed, and the results showed that
$RO_2+NO$ and $OH+NO_2$ dominated local $O_3$ generation and removal, respectively.
This shows that atmospheric free radicals play a key role in localized $O_3$ generation in
Zhengzhou City (Wang et al., 2022). The next subsection describes the free radical
chemistry for an in-depth investigation.
**3.3.2 $HO_x$ budget**
The OH concentration was calculated using the OBM. The simulation results showed
(Fig. S6) that the mean daily peak values of OH and $HO_2$ radicals in Zhengzhou were
$5.6 \times 10^6$ and $3.8 \times 10^8$ molecule/cm$^3$, respectively. A comparison of the results with
those of Tan (2019) showed that the OH concentration in Zhengzhou was slightly
lower than those in Beijing and Shanghai and higher than those in Chongqing and
Guangzhou, However, the concentration of $HO_2$ radicals was lower than that in
Chongqing, consistent with the results of Beijing, Shanghai, and Guangzhou. Previous
experiments based on comprehensive field observations in China have shown that OH
concentrations may be underestimated under low $NO_x$ conditions (Fuchs et al., 2017;
Hofzumahaus et al., 2009). Here, the $NO_x$ concentration was high; thus, the model
reproduced the OH concentration relatively well (Rohrer et al., 2014). The prominent
feature of the high $NO_x$ state is the underestimation of $HO_2$ (Tan et al., 2017), which
has been observed at urban sites outside China (Dusanter et al., 2009; Kanaya et al.,
2007), partially explaining the low $HO_2$ radicals in Zhengzhou City.
Fig. 7 demonstrates the daily trends of HOx radicals under different $O_3$ pollution
conditions. The results showed that the HOx concentration significantly increased



with an increase in photochemical pollution. The average daily peak concentration of
OH radicals on non-polluted days was $4.2 \times 10^6$ molecule/cm$_3$, with peaks increasing
1.5 and 2.7 times under mild and moderate pollution conditions, respectively. For the
HO$_2$ results, the peaks increased 3.6 and 6.4 times, respectively. The aforementioned
phenomena indicate a more active radical cycle during high O$_3$ periods (Zhu et al.,
2020), and the HO$_x$ radical source–sink cycle was investigated.
HO$_X$ radicals trigger VOC oxidation and promote O$_3$ formation. Fig. S7 illustrates the
formation and loss pathways of OH radicals during the observation period. For OH,
the generation pathways are mainly HO$_2$ + NO and O$_3$ + VOCs, with 43% and 56%,
respectively. The removal pathways are mainly based on OH + VOC. Although the air
pollution problems are visually extremely similar, the free radical chemistry,
particularly the primary radical sources, significantly varies across different
metropolitan areas. For example, Lanzhou had a higher OH contribution from O$_3$ +
VOCs (32%) (Jia et al., 2018), whereas Wuhan had a higher contribution of HO$_2$ +
NO (Zhu et al., 2020). O$_3$ photolysis is the main source of OH in Nashville (Martinez
et al., 2003). Nitrous acid (HONO) photolysis plays a dominant role in New York City
(Ren et al., 2003), Paris (Michoud et al., 2012), and Wangdu, China (Tan et al., 2017).
Formaldehyde photolysis is an important source of OH in Milan (Alicke et al., 2002).
Fig. S7 shows the simulated average generation and loss rates of OH for the three
periods. The OH formation or loss rate increased with increased photochemical
pollution, implying a higher efficiency of free radical cycling during photochemical
pollution. The situation of the source–sink pathways of OH in different pollution
periods was similar to that in the observation period, and VOCs and NO$_x$ significantly
impacted the HO$_x$ free radical cycling.
HONO is an important source of HO$_X$, playing a crucial role in atmospheric chemistry
(Xue et al., 2014). Considering that we did not measure the HONO mixing ratio, the
results may be underestimated. Thus, the current OBM-MCM results may have
underestimated the daytime HO$_X$ concentration to an extent, and supplemental HONO
is required to better determine the HO$_X$ balance.





### 3.4 O₃–NOₓ–VOC sensitivity

### 3.4.1 VOC/NOₓ ratio

The influence of $O_3$ precursors on $O_3$ formation can be defined as the VOC and $NO_x$ control zones, which are critical in developing effective strategies for reducing regional $O_3$ pollution. The VOC/$NO_x$ ratio has been widely used to determine the state of $O_3$ formation. Generally, at a VOC/$NO_x$ ratio below 10 (ppbC/ppbv), a VOC-sensitive zone is observed. However, when the ratio exceeds 20, it is in a $NO_x$-sensitive state. At a ratio between 10 and 20, the reduction of VOCs and $NO_x$ can effectively reduce $O_3$ concentrations (Hanna et al., 1996; Sillman et al., 1999).

MEM was selected as the study site to investigate the VOCs/$NO_x$ (ppbC/ppbv) during the high $O_3$ hours from 2019 to 2021. As shown in Table S2, the VOCs/$NO_x$ at this site was $6.2 \pm 7.1$ during the observation period. The ratio increased with an increase in the photochemical pollution levels, i.e., $5.9 \pm 7.3$ for non-polluted days, $7.0 \pm 6.6$ for mildly polluted days, and $7.3 \pm 6.7$ for moderately and highly polluted days. In addition, the fraction of $O_3$-polluted days with VOCs/$NO_x$ > 10 increased. The proportions of mildly and moderately/highly polluted days were 15% and 18%, respectively, indicating that the proportion of $O_3$ generation in Zhengzhou subject to the transition zone increased with increasing photochemical pollution (Zhu et al., 2020).

Fig. 8(b) shows the daily trends of the VOC/$NO_x$ ratios for different photochemical pollution periods. The ratios for the three periods exhibited similar daily variations. Higher ratios were observed at midnight (1:00–6:00), after which the ratios rapidly decreased, indicating that $NO_x$ concentrations increased more rapidly than VOCs in terms of the effect of vehicle emissions (Gu et al., 2019). Thereafter, the VOCs/$NO_x$ ratio increased with the $O_3$ concentrations. In the afternoon (12:00–16:00), at a high $O_3$ concentration, the VOC/$NO_x$ was high in all the periods (moderate and high pollution > light pollution > non-pollution). During moderate and high $O_3$ pollution, the ratio of VOC/$NO_x$ exceeded 10, characterized by the transition zone. synergistic emission reduction of VOCs and $NO_x$ to effectively mitigate photochemical pollution. The VOCs/$NO_x$ ratios are only a preliminary determination of $O_3$ sensitivity and were





subsequently validated by the CMAQ model and OBM.

**3.4.2 Relative importance ratio and empirical kinetic modeling approach using the box model**

RIR is a key parameter for determining the relationship between $O_3$ and its precursors.
Thus, RIR values are important for developing science-based $O_3$ pollution control
strategies. Higher positive RIR values indicate that the precursors are more sensitive
to $O_3$ production, whereas substances with negative RIR values play a negative role in
$O_3$ formation (Niu et al., 2024; Zhang et al., 2024). Here, we quantified the RIR
values of $NO_x$, CO, and different fractions of VOCs and further classified
anthropogenic VOCs (AHCs) into aromatic hydrocarbons, olefins, and alkanes to
better understand the effects of different sources on $O_3$.
The city monitoring station was selected as the target site. The acquired observation
data for the high $O_3$ period (May–September) for 2019–2021 were used to determine
the RIR in Zhengzhou (Fig. 8a). The results showed that the RIR of AHCs was larger
during the observation period, indicating that anthropogenic sources significantly
contribute to local $O_3$ generation, and the reduction of anthropogenic sources of VOCs
can effectively mitigate local $O_3$ pollution. The contribution of BVOCs to local $O_3$
generation was high owing to the high reactivity of BVOCs and the higher emission
intensity caused by the high temperature and strong radiation in May–September. The
RIR of CO was low, indicating that the mitigation of $O_3$ pollution through CO
reduction was ineffective. The negative RIR for $NO_x$ indicated that reducing $NO_x$
contrarily promoted $O_3$ production.
Fig. 8a illustrates the distribution of RIR on $O_3$ non-pollution, mild pollution, and
moderate and heavy pollution days. $RIR_{AHC}$ exhibited high values in the three periods;
therefore, VOC control must be strengthened in the region, particularly olefins and
aromatic hydrocarbons during the $O_3$ pollution hours. The RIR values of BVOCs were
high and tended to increase with an increase in pollution. The values of $O_3$
non-pollution, mild pollution, and moderate and heavy pollution days were 0.4, 0.5,
and 0.7, respectively. The $RIR_{NOx}$ values were negative on $O_3$ non-pollution and mild



pollution days, indicating that it was in the VOC control zone at this time. The $RIR_{NOx}$
value became positive (0.4) with an increase in pollution levels. Thus, the synergistic
control of $NO_x$ and VOCs effectively reduced the photochemical pollution on high $O_3$
days.
Owing to the use of the reactivity concept, EKMA can be employed as a standardized
framework for investigating the sensitivity of regional $O_3$ production to VOCs and
$NO_x$ (Liu et al., 2023; Wang et al., 2022). Thus, based on the study period, when the
photochemical pollution was more severe, the pollutant information and values of
meteorological factors from Zhengzhou monitoring stations were inputted into the
OBM. As shown in Fig. 9, the $O_3$ contours show the local maximum concentration of
$O_3$ as a function of the initial $NO_x$ and VOC concentrations. The relationship between
$O_3$ and its precursors was highly nonlinear. At low $NO_x$ concentrations, the $O_3$
concentration increased almost linearly with increasing $NO_x$ concentration. The
increase in the $O_3$ concentration gradually slowed with an increase in the $NO_x$
concentration, reaching a local peak. The line connecting the localized peaks in $O_3$
concentration is called a "ridge" (Fig. 9). The ridge divides the $O_3$ formation into two
photochemical states. Below the ridge is the $NO_x$ control zone, and the VOC control
zone is above the ridge.
Based on online data from the Zhengzhou monitoring station, an EKMA curve was
plotted (Fig. 9), and the results were consistent with the RIR. Zhengzhou was in the
VOC control zone on $O_3$ non-pollution and mild pollution days. The local $O_3$
susceptibility was transformed into the transition zone as pollution increased. This
indicates that the summer $O_3$ pollution in the urban area of Zhengzhou was mainly in
the VOC-sensitive zone, and reducing the VOC concentration facilitated $O_3$ pollution
control. As shown by the slope of the ridge in Fig. 9, the optimal reduction ratio of
VOCs to nitrogen oxides is 2.9:1, and it is recommended not to be lower than 2:1.
**3.4.3 Regional distribution of $O_3$ sensitivity based on CMAQ-DDM**
Based on the DDM method, the sensitivity of $O_3$ to its precursors was assessed across
different regions of Henan Province (Su et al., 2023; Yu et al., 2021). The province



was categorized into various sensitivity regions according to the ratios of $O_3$ precursor
concentration sensitivities (Fig. 10). The 3-year simulation results revealed that the
VOC-sensitive region in Henan Province encompassed the Anyang-Zhengzhou area
along the Taihang Mountains, including three cities north of the Yellow River, and
areas south of the river, such as Zhengzhou, Xuchang, and parts of Luoyang, Kaifeng,
and Luohe. This aligns with conventional knowledge, as these areas experience high
$NO_x$ emissions; thus, VOC concentration is critical in $O_3$ generation (Su et al., 2023).
In Zhengzhou City, subregional sensitivity analysis indicated that all areas fell within
the VOC control zone, suggesting that $O_3$ management during summer should
prioritize VOC control measures. A comparison of sensitivities in 2021 with those
from the previous 2 years showed a northward shift in the VOC-sensitive area. This
was largely attributed to a significant reduction in anthropogenic emissions owing to
heavy precipitation in Henan in July 2021, which led to a 25% decrease in
anthropogenic emissions in the affected areas. Natural VOC emissions from plants
remained largely unaffected, resulting in a marked increase in the VOC/$NO_x$ mass
ratio. Consequently, the $NO_x$ control area in Henan Province shifted northward, and
the VOC-sensitive area decreased relative to July 2021. However, in Zhengzhou City,
the reduction in anthropogenic emissions did not impact VOC sensitivity. This is
because natural VOC emissions from plants were not the dominant factor, even in the
context of significant reductions in anthropogenic emissions in northern Henan.
Therefore, the alterationin the VOC/NOx ratio resulting from the heavy rainfall was
inadequate to alter $O_3$ sensitivity.
**4 Summary and conclusions**
The pollution characteristics of atmospheric VOCs in Zhengzhou City were analyzed
using real-time VOC data from May to September in the period of 2019-2021. The
sources of atmospheric VOCs and $O_3$ were determined using PMF and CMAQ
models. Factors affecting free radical equilibrium were investigated to highlight the
major factors driving local ozone generation. The main conclusions are summarized
as follows:



**(1) Pollution characteristics and source distribution of VOCs and ozone**

The study clarified the pollution characteristics and sources of ozone and its precursor VOCs in Zhengzhou City. The city suffered severe photochemical pollution during the observation period, with ozone concentration exceeding the standard value by 45% and average VOC concentration of 90.3 ± 52.8 µg/m³. Moreover, VOC concentration increased with the enhancement of ozone pollution. Their concentrations during the non-pollution, mildly polluted, and moderately polluted periods were 84.7 ± 51.0, 96.6 ± 53.4 and 105.3 ± 59.4 µg/m³, respectively. The PMF modeling yielded six factors contributing to VOC emissions, namely motor vehicle exhaust, solvent use, industrial emissions, liquefied petroleum gas (LPG)/natural gas (NG), stationary combustion, and biogenic sources, among which motor vehicle exhaust was the largest source of VOCs. As ozone pollution became more intense, the contribution of motor vehicles to VOCs was 30%, 31% and 35%, respectively. Industrial emissions were the second largest source of VOCs, accounting for 21%. Ultimately, the source apportionment of ozone was performed based on the CMAQ model. The results showed that ozone formation in Zhengzhou was mainly attributed to local anthropogenic emissions, with motor vehicle exhaust and industrial emissions being the two largest sources.

**(2) Mechanism and sensitivity of ozone formation**

The local ozone formation rate and its generation and removal pathways were revealed, and the $O_3$-NOx-VOC sensitivity of Zhengzhou City was comprehensively evaluated. According to the OBM model analysis, the average $P(O_3)$ on non-pollution, mildly polluted, and moderately/highly polluted days was 2.0, 4.5, and 6.9 ppbv/h, respectively. The contribution of $RO_2$+NO to local ozone generation was more than 80%, indicating that atmospheric free radicals had a significant effect on local ozone formation. The HOx radical concentration increased 1.5-6.4 times on polluted days compared with non-pollution days. The results obtained by employing VOCs/NOx ratio, RIR, EKMA and DDM methods indicated that Zhengzhou was located in the control zone of VOCs and was shifting to the transition zone with the increase in the intensity of ozone pollution. AHCs contributed greatly to local ozone formation. In



particular, reducing aromatic hydrocarbons and olefins helped to effectively mitigate
ozone pollution. The optimal reduction ratio of VOCs to NOx was determined to be
2.9:1, which is recommended not to be lower than 2:1.
**(3) Scientific contribution and policy implications**
Three years of observational data were combined with advanced modeling techniques,
such as the CMAQ and OBM model in this study to comprehensively explore ozone
pollution dynamics in Zhengzhou City. The significant impact of vehicle emissions on
ozone and its precursors is consistent with the results of other urban studies (Song et
al., 2018; Yu et al., 2022), reinforcing the key role of controlling mobile sources in
mitigating photochemical pollution. While PMF-based VOC source apportionment is
widely used in urban studies (Farhat et al., 2024; Frischmon et al., 2024), the
inclusion of CMAQ in ozone source tracking provides new insights into the role of
anthropogenic emissions in ozone formation.
In addition, this study provided a comprehensive assessment of ozone sensitivity
using multiple diagnostic methods. The findings confirmed that Zhengzhou City was
located in the VOC control zone, consistent with the results of other urban studies (He
et al., 2022; Santiago et al., 2024; Tran et al., 2023; Tudor, 2022). However, notably,
different diagnostic methods used for analyzing ozone formation sensitivity have their
inherent advantages and limitations. To accurately determine the $O_3$-NOx-VOC
sensitivity, high-resolution observations were combined with multiple methods to
ensure reliable, scientifically sound conclusions that help to identify key factors
controlling local ozone formation and provide actionable insights for mitigating
photochemical pollution.
In this study, the ozone sensitivity was evaluated using multiple methods, resulting in
a reliable understanding of the complex interactions between $O_3$, NOx, and VOCs,
emphasizing the need for balanced emission control strategies. The proposed optimal
VOC/NOx reduction ratio was identified to be 2.9:1, which provides a practical
framework for the effective control of ozone pollution. This approach addresses the
limitations of single-pollutant abatement measures and ensures that emission control
policies are both scientific and practically feasible.



The results of this study are particularly relevant to rapidly urbanizing regions such as
Zhengzhou, where industrialization and motorization are driving significant changes
in air quality. By highlighting the importance of controlling transportation emissions
and optimizing the VOC/NOx emission reduction ratio, this study provides a solid
scientific basis for air quality management. And this study also emphasizes the need
for integrated emission control strategies that take into account the unique sources and
interactions of pollutants in such environments. These insights are of guiding
reference for the development of targeted policies to address photochemical pollution,
ultimately contributing to the long-term improvement of air quality in rapidly growing
urban areas.
**(4) Limitations and future research directions**
This study has certain limitations. The accuracy of the OBM model depends on
high-quality input data, and the lack of measured HONO data potentially lead to
underestimated HOx radical concentration. Future studies are advised to incorporate
measured HONO data and explore advanced techniques such as machine learning to
improve data quality and reduce model uncertainty. In addition, long-term monitoring
and modeling efforts are required to capture seasonal and inter-annual variations in
ozone formation mechanism.
**Author contributions**
YSJ: Writing-original draft, Methodology, Data curation, Investigation, Visualization,
Validation, Software, Formal analysis. LHY: Formal analysis, Data curation,
Investigation. WH: Supervision, Resources, Project administration, Funding
acquisition. SFC: Methodology,Software. WBB: Data curation, Supervision. YMH:
Data curation, Resources. SKA, WZX, XDQ: Data curation,Validation. ZRQ: Writing
– review & editing, Supervision, Resources, Project administration, Funding
acquisition.
**Competing Interests**
The contact author has declared that none of the authors has any competing interests.



**Acknowledgments**
This work was supported by Key Research and Development Program of Henan (No.
241111320300) and National Key Research and Development Program of China (No.
2017YFC0212403).
**Data availability**
The data set is available to the public and can be accessed upon request from Ruiqin
Zhang (rqzhang@zzu.edu.cn).

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







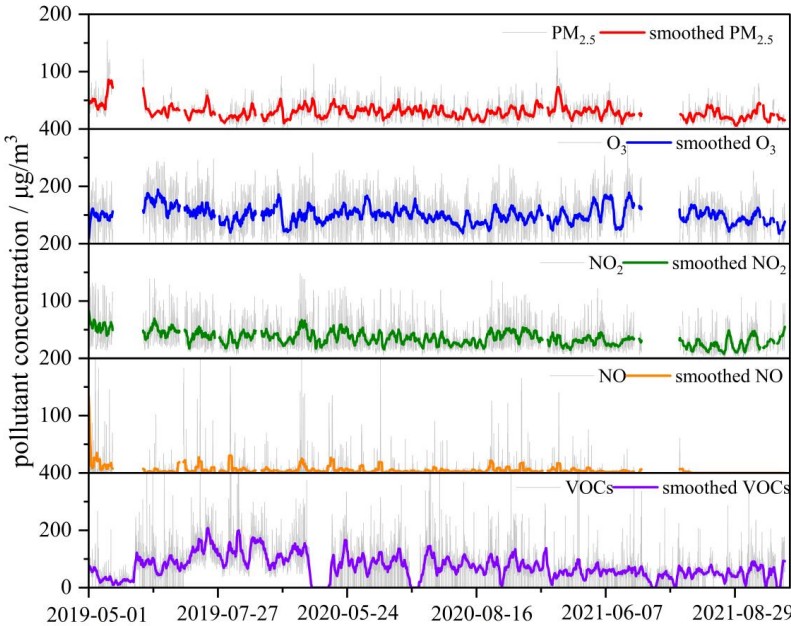

**Fig.1** Smoothing the Time Series of Pollutants. Savitzky-Golay smoothing denoising method was employed to facilitate a clearer and more intuitive observation of pollutant trends, with the window size set to 50 points and the polynomial order configured to 1.






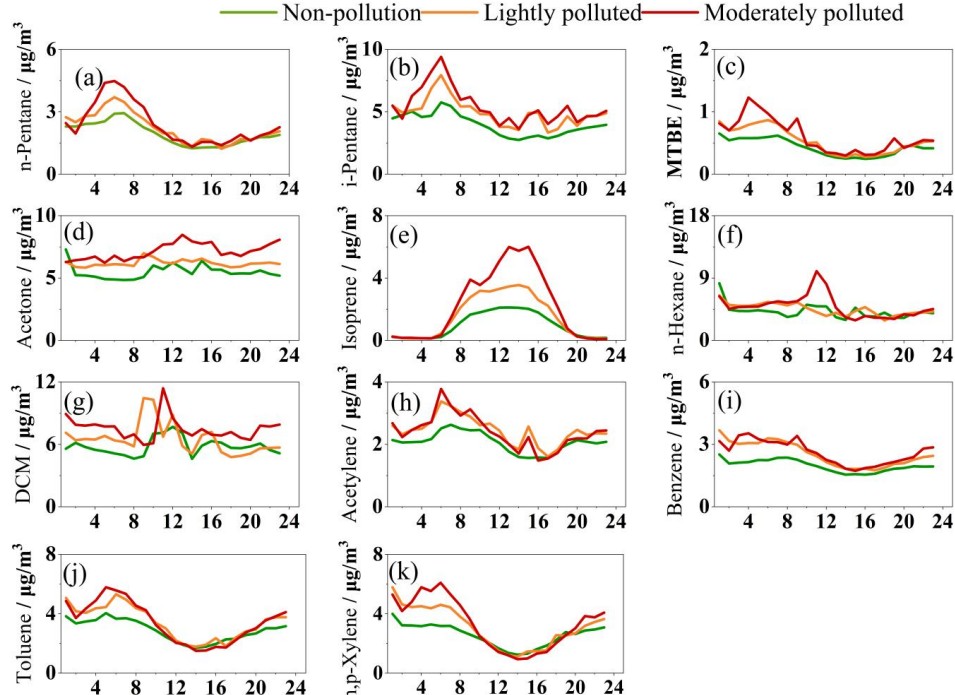


**Fig. 2** Diurnal variations in concentrations of some reactive VOC species in
Zhengzhou under different pollution levels.




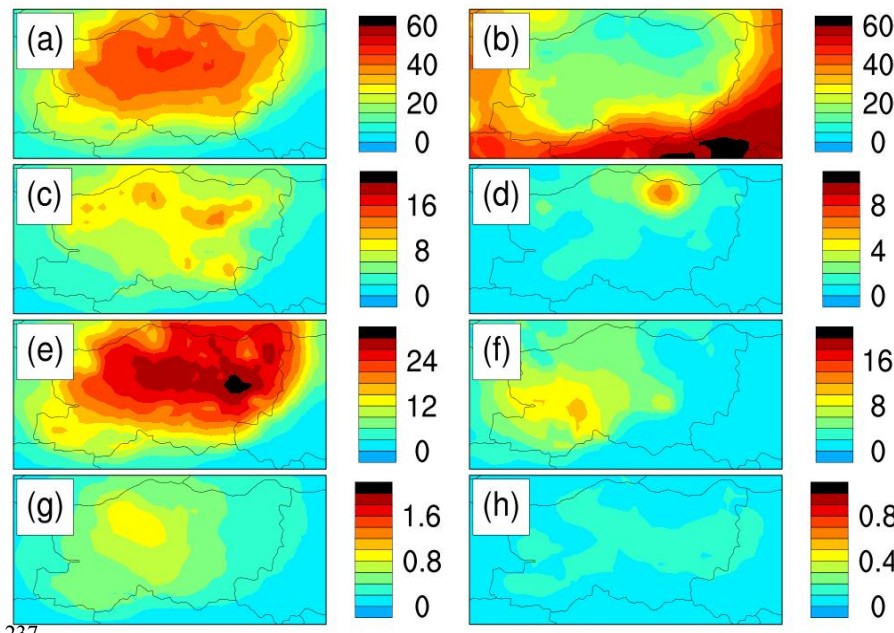


**Fig. 3** O$_3$ sector contribution distribution in Zhengzhou:(a) local anthropogenic
emissions in Zhengzhou, (b) anthropogenic contributions from other regions in Henan
Province, (c) industry, (d) solvents, (e) transportation, (f) electricity, (g) residential, (h)
others.







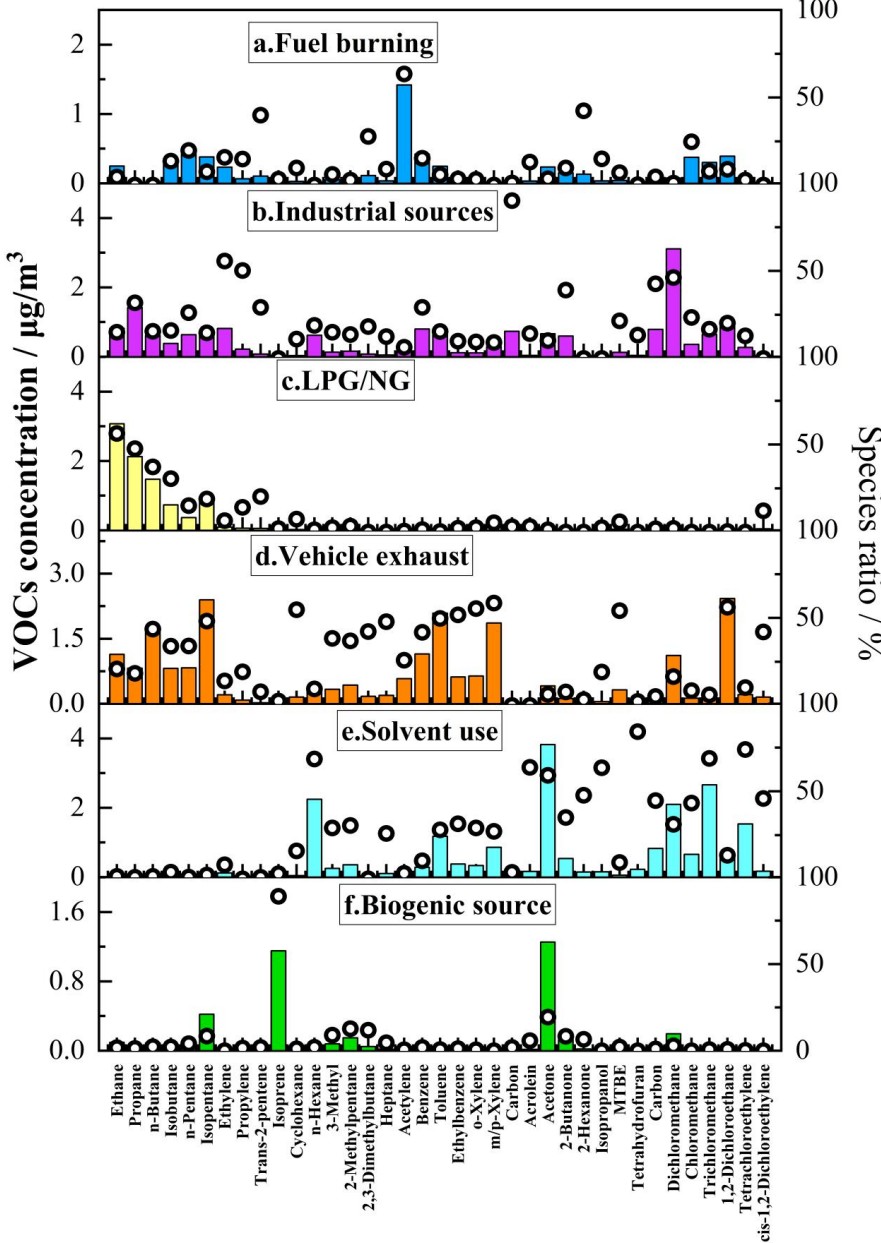


**Fig. 4** Source profiles of six factors derived from PMF modeling.







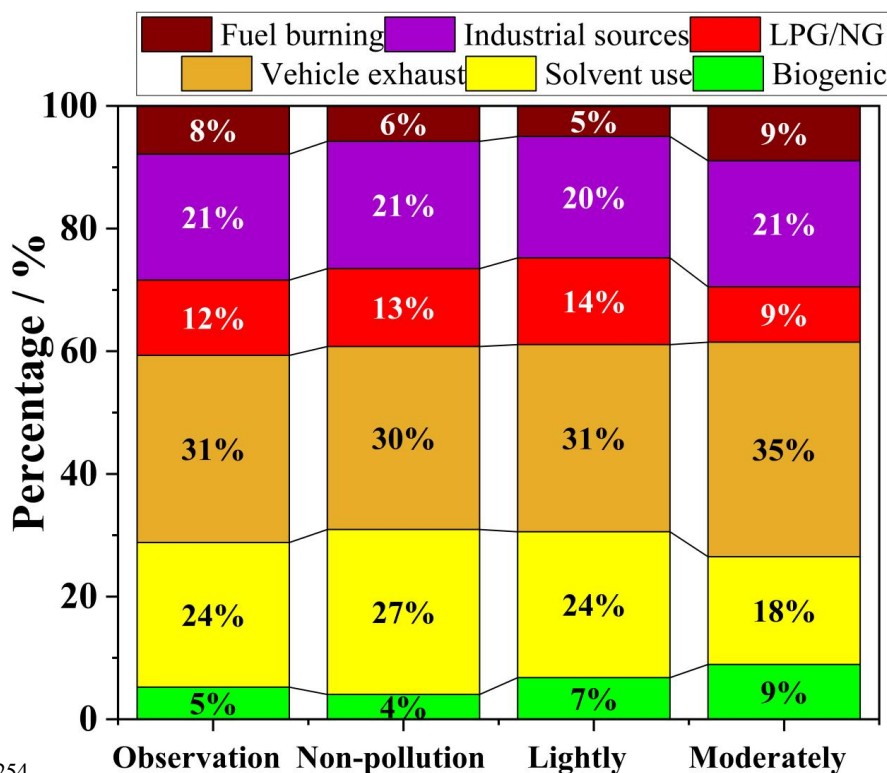

**Fig. 5** Contributions (%) for the six sources identified by PMF model during the sampling period.






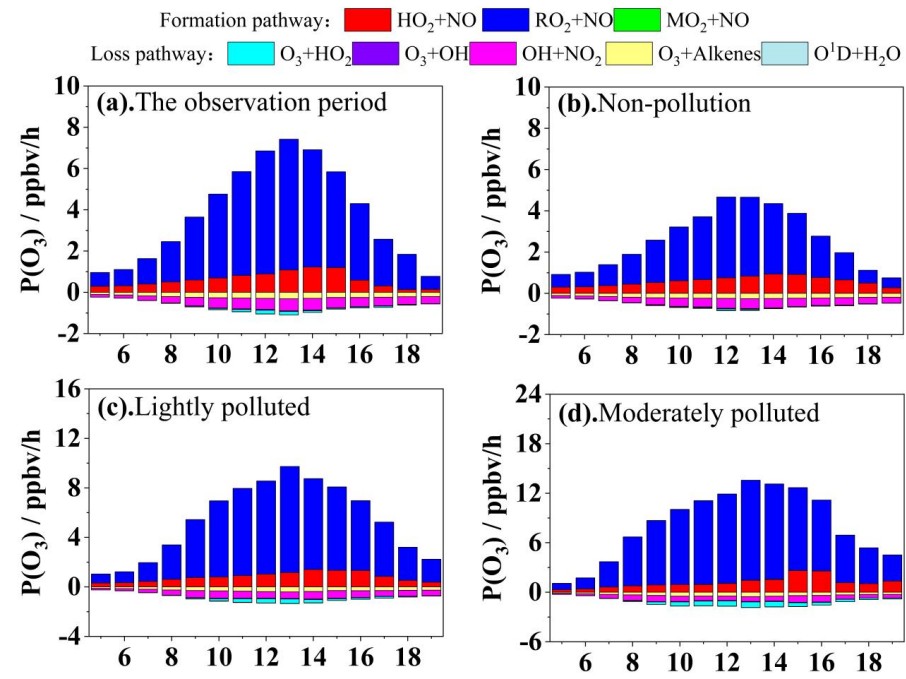

1260         **Fig. 6** Diurnal patterns of $O_3$ production and destruction rates under different

1261                         pollution levels.







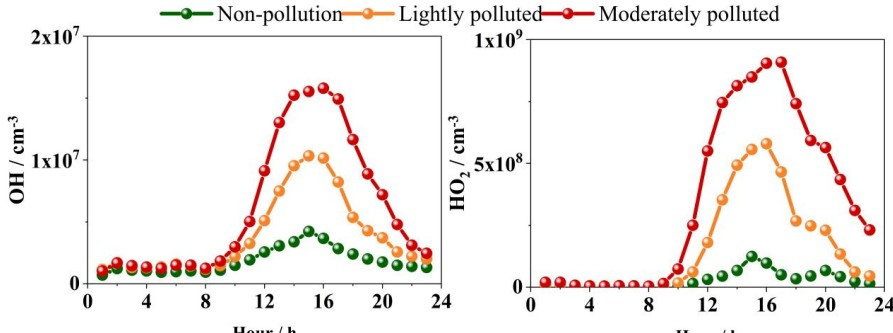


**Fig. 7** Diurnal variation distribution of HOx radicals under different pollution levels.




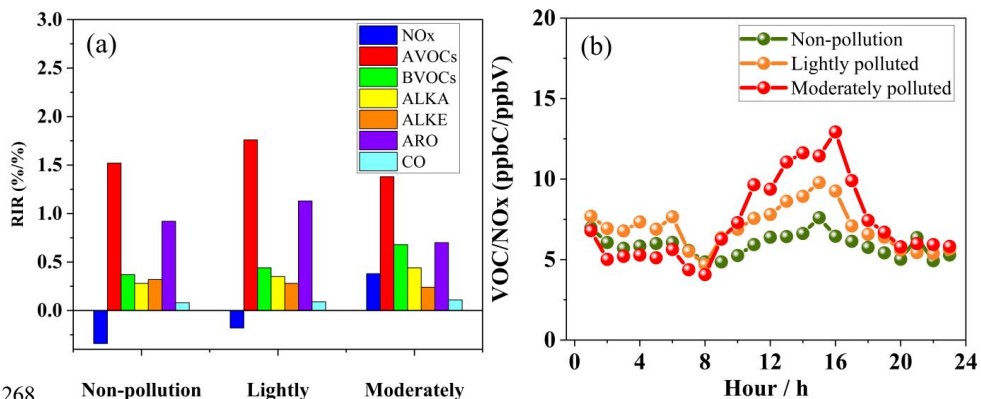

**Fig. 8** Distribution of O$_3$-NOx-VOC sensitivity in Zhengzhou under different
pollution levels.





1273
1274

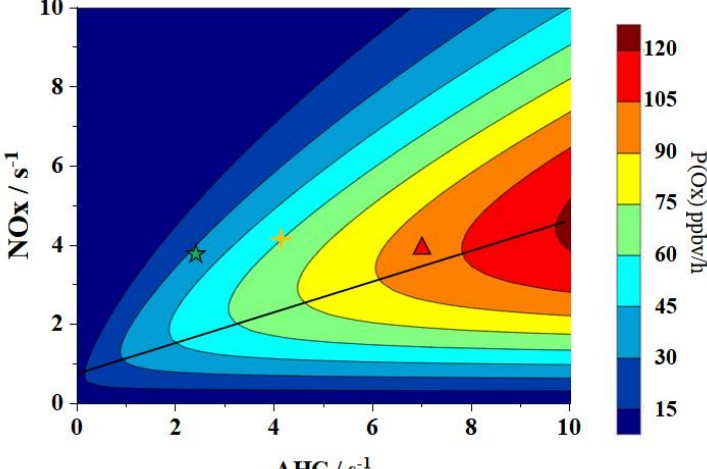

1275

**Fig. 9** The nonlinear relationship between the local ozone production rate and the
activities of VOCs and NOx under different pollution levels. The x-axis represents the
OH reaction activity of AHCs, while the y-axis represents the OH reaction activity of
NOx. The black straight line indicates the ridge line, and the black contour lines
represent the local ozone production rate, measured in ppbv/h. The green pentagram,
orange four-pointed star, and red triangle correspond to non-pollution days, mild
pollution days, and moderate pollution days, respectively.



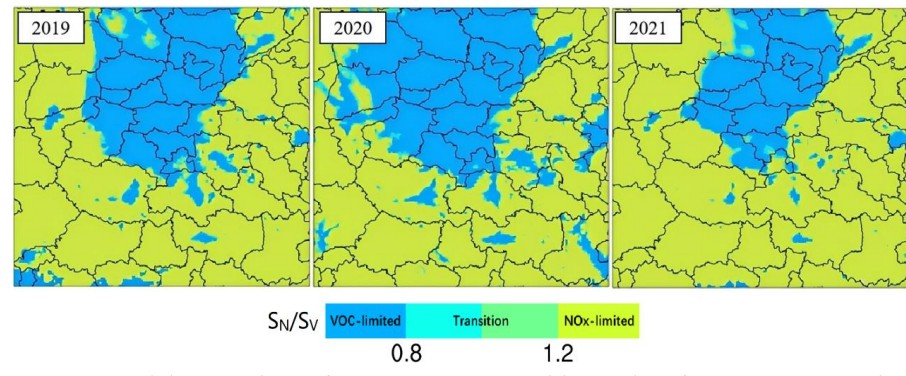

**Fig. 10** Spatial comparison of O$_3$-NOx-VOCs sensitive regime from 2019 to 2021 in Zhengzhou.



**Table list:**

**Table 1** Meteorological factors and pollutant concentrations under different pollution levels.

**Table 2** Concentrations and standard deviations of the Top 20 VOCs by concentration during different pollution periods(μg/m$^3$).

**Table 3** Contribution proportions of $O_3$ formation and removal pathways during the observation Period (%).



1298        **Table 1** Meteorological factors and pollutant concentrations under different pollution levels.

|  | Unit | Non-pollution | Lightly | Moderately | Average |
|---|---|---|---|---|---|
| WS | m/s | 1.6±1.1 | 1.8±1.2 | 1.7±1.1 | 1.7±1.2 |
| RH | % | 74.9±37.6 | 62.6±30 | 56.7±26.6 | 69.4±34.8 |
| T | °C | 25±11.2 | 27.9±11.4 | 29.6±12.1 | 26.3±11.5 |
| VOCs | µg/m$^3$ | 84.7±51 | 96.6±53.4 | 105.3±59.4 | 90.3±52.8 |
| NO | µg/m$^3$ | 5.7±18.3 | 6.2±18.1 | 5.3±13 | 5.8±17.9 |
| NO$_2$ | µg/m$^3$ | 32.7±25.6 | 38.7±28.8 | 40.7±25.9 | 35.3±27 |
| PM$_{10}$ | µg/m$^3$ | 69.8±61.9 | 88.3±58.1 | 100.1±54.5 | 78.3±61.1 |
| PM$_{2.5}$ | µg/m$^3$ | 27.1±20.9 | 32.8±20.4 | 35.5±16.4 | 29.7±20.7 |




**Table 2** Concentrations and standard deviations of the Top 20 VOCs by concentration during
different pollution periods(μg/m$^3$).

| Non-pollution | Ave ± SD | Lightly | Ave ± SD | Moderately | Ave ± SD |
|---|---|---|---|---|---|
| Dichloromethane | 6.7 ± 12.6 | Dichloromethane | 6.4 ± 15.6 | Dichloromethane | 7.6 ± 7.7 |
| Ethane | 5.5 ± 4 | Acetone | 6.2 ± 4.5 | Acetone | 7.2 ± 4.7 |
| Acetone | 5.1 ± 5.9 | Ethane | 5.4 ± 2.8 | n-Hexane | 6 ± 27.3 |
| Propane | 4.4 ± 3.2 | Isopentane | 4.9 ± 4.7 | Ethane | 5.5 ± 2.9 |
| n-Hexane | 4.1 ± 10.6 | n-Hexane | 4.5 ± 7.7 | Isopentane | 5.4 ± 4 |
| Isopentane | 4 ± 3.6 | Propane | 4.1 ± 3.5 | 1,2-Dichloroethane | 4 ± 3.2 |
| n-Butane | 4 ± 3.5 | n-Butane | 3.8 ± 3.4 | Propane | 3.7 ± 2.2 |
| 1,2-Dichloroethane | 3.6 ± 5.2 | 1,2-Dichloroethane | 3.7 ± 3.4 | Toluene | 3.4 ± 2.9 |
| Toluene | 3.5 ± 3.7 | Toluene | 3.4 ± 3.5 | n-Butane | 3.3 ± 2.5 |
| m/p-Xylene | 3.2 ± 4.3 | m/p-Xylene | 3.1 ± 4 | m/p-Xylene | 3.3 ± 3.7 |
| Trichloromethane | 2.9 ± 5.3 | Trichloromethane | 2.8 ± 3.2 | Trichloromethane | 2.8 ± 1.8 |
| Naphthalene | 2.4 ± 4.7 | Tetrachloroethylene | 2.7 ± 3.6 | Benzene | 2.6 ± 1.8 |
| Benzene | 2.3 ± 1.7 | Acetylene | 2.5 ± 3 | Tetrachloroethylene | 2.6 ± 3 |
| Acetylene | 2.3 ± 1.9 | Benzene | 2.5 ± 2 | n-Pentane | 2.4 ± 1.9 |
| Isobutane | 2.2 ± 1.9 | n-Pentane | 2.3 ± 2 | Acetylene | 2.4 ± 1.5 |
| n-Pentane | 2.1 ± 1.7 | Vinyl acetate | 2.2 ± 3.3 | Isoprene | 2.3 ± 3.1 |
| Ethylene | 1.8 ± 1.2 | Isobutane | 2.2 ± 1.6 | Vinyl acetate | 2.2 ± 3.1 |
| Tetrachloroethylene | 1.8 ± 3.3 | Carbon tetrachloride | 1.8 ± 2.3 | Isobutane | 2.1 ± 1.1 |
| Vinyl acetate | 1.6 ± 4.1 | Freon 12 | 1.6 ± 1.8 | 2-Methylpentane | 1.9 ± 2.6 |
| Freon 11 | 1.6 ± 0.7 | Ethylene | 1.6 ± 1.4 | 2-Butanone | 1.8 ± 1 |




**Table 3** Contribution proportions of $O_3$ formation and removal pathways during the observation
Period (%).

|  | The observation period | Non-pollution | Lightly | Moderately |
|---|---|---|---|---|
| $HO_2+NO$ | 16 | 23 | 15 | 14 |
| $MO_2+NO$ | <1 | <1 | <1 | <1 |
| $RO_2+NO$ | 84 | 77 | 85 | 85 |
| $O_3+alkenes$ | 33 | 35 | 31 | 28 |
| $OH+NO_2$ | 56 | 58 | 51 | 43 |
| $O_3+OH$ | 2 | 2 | 3 | 3 |
| $O_3+HO_2$ | 8 | 4 | 15 | 25 |
| $O^1D+H_2O$ | <1 | 1 | 1 | 1 |

