# Peer review of "Diagnosing $O_3$ formation and $O_3$-$NO_X$-VOC sensitivity in a heavily polluted megacity of central China: A multi-method systematic evaluation over the warm seasons from 2019 to 2021"

_EGUsphere, 2024_

## Author Comment (AC1)

**Manuscript number:** EGUSPHERE-2024-4178

**Title:** Diagnosing $O_3$ formation and $O_3$-NOX-VOC sensitivity in a heavily polluted megacity of central China: A multi-method systematic evaluation over the warm seasons from 2019 to 2021

Yu et al. analyzed the two-year routine hourly measurement data from a municipal environmental monitoring station in Zhengzhou City, China. Using PMF and CMAQ, they found that O3 formation was primarily driven by VOC emissions from the transportation sectors. Furthermore, they deployed an OBM model to provide insights into the equilibrium of free radicals. Based on the analysis, the authors claim that the O3 formation is limited by VOCs.

The current version is long to read and poorly written. Some terminologies are badly defined or not consistent throughout the manuscript. The PMF analysis is not technically sound, and the study reads like a measurement report overall. In addition, the comparison between studies is limited to other Chinese studies. I don't think readers from countries other than China could benefit from the findings. Considering the work lacks a broad readership, I suggest rejecting the manuscript.

**Response:** Thank you for your careful reading of our paper and the valuable comments and constructive suggestions. Below are the point-to-point responses to all the comments (The comments are marked in black font and the responses are marked in dark blue font). The major changes that have been made according to these responses were marked in yellow color in the highlighted copy of the revised manuscript. And our own minor changes were marked in red font. Note that the following line numbers are shown in the corrected version.

**Major Comments:**

1. Introduction is long and reads very descriptive. It is hard to see the novelty and importance of the work. I will suggest the authors make it compact and concise and, at the same time, highlight the novelty and importance of the study.

**Response:** We sincerely appreciate the reviewer's insightful feedback. In response to your suggestions, we have made comprehensive revisions to the Introduction section, which include the following improvements:

Conciseness: The original text has been streamlined from 1,100 to 850 words, with professional language editing services applied to enhance clarity and readability.

Global Perspective: We have broadened the international scope by systematically comparing the progress of ozone pollution mechanism research in both domestic (China) and international contexts (e.g., the EU, and USA). This revision better situates our work within the global research landscape.

Three key research gaps are identified: (1) Most observational studies focus on short-term pollution events and lack continuous tracking of VOC compositional dynamics across different ozone levels. (2) While VOC source apportionment is well-explored, ozone source attribution remains insufficiently addressed. (3) The limitations of single-method approaches for identifying ozone sensitivity emphasize the need for integrated multi-method frameworks.

These revisions collectively sharpen the focus on the innovative contributions of our study. First, it conducts a comprehensive analysis of the pollution characteristics and formation mechanisms of ozone and VOCs based on three years of data. This fills the gap in current research, which tends to focus on short-term pollution events, by providing continuous, dynamic information on pollution trends. Second, the study utilizes the CMAQ and PMF models to conduct source apportionment of ozone and its precursor VOCs. This multi-model approach allows for a more detailed identification of pollution sources, enhancing the accuracy of source analysis. Finally, the study employs a multi-method approach to assess ozone sensitivity, overcoming the limitations of using a single method. By integrating different methods, the study offers a more comprehensive and accurate evaluation of ozone sensitivity. These innovations contribute to a deeper understanding of ozone pollution and its formation mechanisms, while also providing valuable insights for refining ozone.

2.  Lines 316 – 324: The way how VOC species were included in the PMF analysis seems subjective. How did the author define the high frequency of concentrations below MDL? What are local emissions? How do they contribute to abnormally high concentrations? Which VOC compounds had abnormally high concentrations? Do they always exhibit abnormally high concentrations? Why is the signal-to-noise threshold defined as 5?

**Response:** Thank you very much for your valuable comments. I would like to provide a detailed response to your queries as follows:

Regarding the selection of VOC species for the PMF analysis, we followed several principles based on the manual published by the EPA to ensure the scientific rigor and validity of our data:(1)We prioritized VOC species with relatively high concentrations to ensure that their signals could be effectively separated during the analysis.(2)We selected species that are commonly recognized as emission tracers (e.g., isoprene) to help accurately trace the pollution sources.(3)We excluded species with more than 25% missing data or concentrations below the method detection limit (MDL) to avoid analysis bias caused by incomplete or very low data.(4)Species with a signal-to-noise ratio (S/N) lower than 0.5 were also excluded to ensure that only species with sufficient signal strength were included in the analysis. I have updated the relevant content in section 2.3.2, as shown below.

The species selection of PMF is performed based on the following principles: (1) prioritizing VOC species with higher concentrations for effective signal separation; (2) selecting emission tracers like isoprene to accurately trace pollution sources; (3) excluding species with over 25% missing data or concentrations below the method detection limit (MDL) to avoid bias; (4) excluding species with a signal-to-noise ratio (S/N) below 0.5 to ensure adequate signal strength.

Regarding the high concentration issue mentioned in the manuscript, we observed that there were occasional spikes in concentrations of compounds such as trichloromethane. After further investigation, we concluded that these anomalies were due to specific laboratory emissions. Therefore, these outliers were removed during the data cleaning process and were not included in the overall analysis of the paper. I also apologize for the confusion regarding the signal-to-noise threshold. In the original manuscript, we mistakenly mentioned the threshold as 5, but it should have been 0.5. We have corrected this in the revised version.

3. Lines 325 – 332: The current description for selecting PMF factor solution is vague. Why was Q/Qexp not chosen to find the optimal factor solution? This is a well-established variable for searching for the factor solution. The ideal

solution should provide a Q/Qexp close to 1.0. The change in Q/Qexp as a function of the number of factors should be presented as a figure in the Supplement. In addition, no uncertainty estimation method (e.g., bootstrap, displacement) is mentioned to examine the solution's robustness. Is the chosen PMF solution for presentation robust enough?

**Response:** Thank you for your valuable comments on our manuscript, and we appreciate the opportunity to address your concerns.

Regarding the selection of the PMF factor solution, we identified a six-factor solution based on the method described by Ulbricht et al. (2009). Specifically, we utilized two key parameters to determine the optimal number of factors: (1) Qtrue/Qrobust values and (2) Qtrue/Qtheoretical values. These parameters have been successfully applied in previous studies, including one of my own published works, to assess the appropriateness of the chosen factor solution. We will include a figure in the Supplement that demonstrates the change in Q/Qexp as a function of the number of factors, as this can offer additional clarity to the decision-making process.

[Figure]

**Fig. S3** The ratios of Qtrue/Qrobust and Qtrue/Qexp at factor size ranged from 3–7.

Thank you very much for your valuable comments, and we have added uncertainty

estimation methods (bootstrap method and displacement method). As shown in Table S3, the stability assessment of the six-factor solution of PMF by Bootstrap resampling (BS) and parametric displacement test (DISP) shows that the solution has high statistical robustness.

Table S3 Evaluating the robustness of the solution using bootstrap and displacement tests

| Diagnostics | 3 factors | 4 factors | 5 factors | 6 factors | 7 factors |
| --- | --- | --- | --- | --- | --- |
| DISP % dQ | 0 | 0 | 0 | 0 | 0 |
| DISP % swaps | 0 | 0 | 0 | 0 | 0 |
| BS Mapping <80% | | | Factor 5 (62%) | | Factor 5 (69%) |
| BS-DISP | Successful | Successful | Successful | Successful | Successful |

4. Lines 336 – 346: Several terminologies are badly defined. What concentrations are considered high O3 values? What concentration is the national secondary standard limit? This needs to be highlighted as a horizontal line in Figure 1. What is defined as moderate, high, or severe pollution days? Is it based on the concentration of O3 or PM2.5? Was the downward trend statistically significant?

**Response:** This study focuses on the period from May to September, during which ozone pollution frequently occurs. The high ozone value zone in this study is defined as the number of days with ozone levels exceeding the standard, specifically when the 8-hour average ozone concentration (MDA8) exceeds 160 μg/m³, which is above the national secondary standard (GB3095-2012). Figure 1 has been updated to reflect this, and a horizontal line has been added to highlight the trend of ozone concentration. Regarding the classification of pollution levels, the standards are as follows:Mild pollution: MDA8 > 160 μg/m³; Moderate pollution: MDA8 > 215 μg/m³; Severe pollution: MDA8 > 265 μg/m³. These thresholds are based on ozone concentration, not PM2.5 levels. Moreover, the downward trend in ozone concentration observed in the study was statistically significant. To avoid causing confusion for readers, we have revised this section and updated Figure 1 in accordance with the reviewer's suggestions.

The results indicate severe photochemical pollution in the period from May to September during the years 2019-2021 (in Fig. 1). According to the GB 3095-2012 standard, the maximum daily average 8 h (MDA8) $O_3$ concentrations exceeding 160 160 μg/m³ and 160 μg/m³ are categorized as light pollution and moderate pollution,

respectively. The proportion of days with O$_3$ concentrations exceeding 160 µg/m³ was as high as 45%, with days classified as moderate or higher pollution accounting for 7%. The MDA8 was recorded on June 6, 2021 (285 µg/m³), with a severe pollution level. The proportion of MDA8 O$_3$ concentrations exceeding the standard during sampling periods from 2019 to 2021 was 53%, 37%, and 36%, showing a downward trend but still indicating severe O$_3$ pollution.

[Figure]

**Fig.1** Smoothing the Time Series of Pollutants. Savitzky-Golay smoothing denoising method was employed to facilitate a clearer and more intuitive observation of pollutant trends, with the window size set to 50 points and the polynomial order configured to 1.

5. Lines 347 – 363: The Pearson correlation coefficient is a correlation coefficient that measures the linear correlation between two variables. Therefore, a low Pearson correlation coefficient indicates that the linear correlation is weak, but it does not necessarily imply that the non-linear correlation is also weak. In addition, the correlation between O3 and its precursors only makes sense if the meteorological conditions have been normalized. I highly recommend the authors choose a proper statistical method to analyze the relationships between variables.

**Response:** First of all, we would like to sincerely thank you for your valuable feedback. We fully understand that the Pearson correlation coefficient primarily measures the linear relationship between two variables, and that a lower Pearson

correlation does not necessarily imply no relationship between the variables; there may be a non-linear relationship. Therefore, in the revised version of our manuscript, we will include a discussion on the limitations of the Pearson correlation coefficient and consider using other statistical methods (such as the Spearman rank correlation or Kendall's tau) to assess potential non-linear relationships. We believe this will contribute to a more comprehensive understanding of the relationships between variables. The relevant details will be added in Appendix Tables S4 and S5. It is worth noting that the results obtained using these methods are consistent with those of the Pearson correlation, further enhancing the robustness of our analysis.

Regarding the correlation between $O_3$ and its precursors, we agree with your suggestion that meteorological conditions should be taken into account in the analysis. Based on this, we will emphasize the importance of meteorological conditions in the analysis and mention. In another paper we are currently working on, machine learning methods (such as XGBoost and Random Forest) have been used to identify the contribution of meteorological factors to $O_3$ formation. Our research shows that meteorological conditions account for 58.3% of the total contribution to $O_3$ formation, which provides an important basis for further refining the analysis in this study.

**Minor Comments:**

1. Lines 73 – 79: The authors listed two studies from Taichung and Wuhan as examples. Readers not familiar with Chinese geography have no clue about these two cities. Proper descriptions need to be provided for these two example cities. Lines 91 – 93: Some descriptions need to be provided for Henan here.

**Response:** Thanks for your valuable suggestions. The introduction section has been rewritten.

2. Line 100: What are the transition zones?

**Response:** The transition zone in ozone sensitivity refers to the region where ozone production is highly responsive to changes in both volatile organic compounds (VOCs) and nitrogen oxides (NOx). This zone lies between the VOCs-dominated and

NOx-dominated areas, where the ratio of VOCs to NOx approaches a critical threshold for photochemical reactions. Within this zone, ozone production is influenced by the combined effects of VOCs and NOx, exhibiting a nonlinear response. For instance, reducing NOx alone can sometimes lead to an increase in ozone levels. Similarly, reducing VOCs alone may only result in a modest reduction in ozone concentrations, due to the inhibition of free radical chain reactions.

3. Lines 171 – 172: How often was the particulate removal device cleaned? I am concerned that during high pollution periods in China, the removal device can be saturated with high aerosol mass loading very quickly.

**Response:** Thank you for your comment. As per the HJ1012-2018 technical standards and the equipment manual, the particulate removal device, including the PM2.5 cutter and sampling head, is cleaned at least once a month. During high pollution events, such as sandstorms or periods when PM2.5 concentrations exceed 300μg/m³, cleaning is performed within 24 hours after the pollution event to prevent particle deposition and blockage in the system. Additionally, the online monitoring system (TH-300B) incorporates advanced features such as dynamic pulse backflushing and hydrophobic coating to mitigate the impact of high aerosol mass loading and maintain operational efficiency during periods of high pollution.

4. Section 2.1: How many VOC species were identified?

**Response:** Thank you for your valuable comment. In Section 2.1, we analyzed 108 VOC species using both the TO-15 and PAMS methods. The TO-15 method (U.S. EPA Method TO-15) is designed to target 65 non-polar and weakly polar toxic VOCs, such as halogenated hydrocarbons and light alkanes, by utilizing canister sampling and GC-MS. The PAMS method, on the other hand, focuses on 56 photochemically reactive organic compounds which are key to ozone and secondary organic aerosol formation. These methods were selected to comprehensively capture a wide range of VOC species relevant to air quality monitoring.

5. Lines 367 – 369: "Nighttime short-term… the next day (Du et al., 2024)." are redundant.

**Response:** Thank you for your comment. We appreciate your feedback. As suggested, we have removed the redundant sentence "Nighttime short-term… the next day (Du et al., 2024)" to avoid repetition and improve the clarity of the manuscript.

6. Table 1: How many hours or days are defined as non-pollution, lightly pollution, and moderately pollution periods? What are the definitions of non-pollution, lightly pollution, and moderately pollution periods?

**Response:** Thank you for your comment. According to the "Ambient Air Quality Standards" (GB 3095-2012), the thresholds for ozone pollution levels are defined as follows: mild pollution, moderate pollution, and severe pollution based on the MDA8 (Maximum Daily 8-Hour Average) concentrations of 160 μg/m³, 215 μg/m³, and 265 μg/m³, respectively. In our study, the non-pollution period is defined as MDA8 concentrations below 160 μg/m³, the lightly polluted period as MDA8 concentrations between 160 μg/m³ and 215 μg/m³, and the moderately polluted period as MDA8 concentrations between 215 μg/m³ and 265 μg/m³. These definitions align with the national standards for categorizing ozone pollution levels.

The content of Table 1 has been updated, with the proportions of days categorized as non-polluted, light pollution, and moderate or higher pollution being 58%, 45%, and 7%, respectively.

7. Lines 374 – 376: "The daily average… non-pollution days". Is it the difference statistically significant?

**Response:** I apologize for any inconvenience caused. The purpose of this statement is to describe that, daily average $PM_{2.5}$ concentrations on ozone-polluted days exceeded those on non-polluted days.

8. Lines 382 – 383: How did the author determine the atmospheric oxidation capacity and free radical concentrations? I find the statement is very speculative.

**Response:** Sorry for the mistake. We have revised the statement as follows: This may be attributed to the fact that, under $O_3$-polluted conditions, atmospheric oxidation capacity could potentially be enhanced, with possibly elevated concentrations of both O3 and free radicals that might accelerate NO consumption.

9. Lines 386 – 387: What is the contribution of the top 20 substances to the total VOC concentrations?

**Response:** Thank you for your valuable feedback. The feedback has been used to update the content of Table 2. The top 20 substances account for 68.9% to 76.9%.

| Non-pollution | Ave ± SD | Lightly | Ave ± SD | Moderately | Ave ± SD |
|---|---|---|---|---|---|
| Dichloromethane | 6.7 ± 12.6 | Dichloromethane | 6.4 ± 15.6 | Dichloromethane | 7.6 ± 7.7 |
| Ethane | 5.5 ± 4 | Acetone | 6.2 ± 4.5 | Acetone | 7.2 ± 4.7 |
| Acetone | 5.1 ± 5.9 | Ethane | 5.4 ± 2.8 | n-Hexane | 6 ± 27.3 |
| Propane | 4.4 ± 3.2 | Isopentane | 4.9 ± 4.7 | Ethane | 5.5 ± 2.9 |

| | | | | | |
|---|---|---|---|---|---|
| n-Hexane | 4.1 ± 10.6 | n-Hexane | 4.5 ± 7.7 | Isopentane | 5.4 ± 4 |
| Isopentane | 4 ± 3.6 | Propane | 4.1 ± 3.5 | 1,2-Dichloroethane | 4 ± 3.2 |
| n-Butane | 4 ± 3.5 | n-Butane | 3.8 ± 3.4 | Propane | 3.7 ± 2.2 |
| 1,2-Dichloroethane | 3.6 ± 5.2 | 1,2-Dichloroethane | 3.7 ± 3.4 | Toluene | 3.4 ± 2.9 |
| Toluene | 3.5 ± 3.7 | Toluene | 3.4 ± 3.5 | n-Butane | 3.3 ± 2.5 |
| m/p-Xylene | 3.2 ± 4.3 | m/p-Xylene | 3.1 ± 4 | m/p-Xylene | 3.3 ± 3.7 |
| Trichloromethane | 2.9 ± 5.3 | Trichloromethane | 2.8 ± 3.2 | Trichloromethane | 2.8 ± 1.8 |
| Naphthalene | 2.4 ± 4.7 | Tetrachloroethylene | 2.7 ± 3.6 | Benzene | 2.6 ± 1.8 |
| Benzene | 2.3 ± 1.7 | Acetylene | 2.5 ± 3 | Tetrachloroethylene | 2.6 ± 3 |
| Acetylene | 2.3 ± 1.9 | Benzene | 2.5 ± 2 | n-Pentane | 2.4 ± 1.9 |
| Isobutane | 2.2 ± 1.9 | n-Pentane | 2.3 ± 2 | Acetylene | 2.4 ± 1.5 |
| n-Pentane | 2.1 ± 1.7 | Vinyl acetate | 2.2 ± 3.3 | Isoprene | 2.3 ± 3.1 |
| Ethylene | 1.8 ± 1.2 | Isobutane | 2.2 ± 1.6 | Vinyl acetate | 2.2 ± 3.1 |
| Tetrachloroethylene | 1.8 ± 3.3 | Carbon tetrachloride | 1.8 ± 2.3 | Isobutane | 2.1 ± 1.1 |
| Vinyl acetate | 1.6 ± 4.1 | Freon 12 | 1.6 ± 1.8 | 2-Methylpentane | 1.9 ± 2.6 |
| Freon 11 | 1.6 ± 0.7 | Ethylene | 1.6 ± 1.4 | 2-Butanone | 1.8 ± 1 |
| Proportion of top 20 | 76.9% | | 70.1% | | 68.95% |

**Response:** Thank you for your valuable feedback. Table 2 has been updated.

10. Lines 402 – 405: Could you label P1, P2, and P3 in Figure S5? Same for Figure 2.

**Response:** Thank you for your suggestion. The Figure S5 and Figure 2 have been updated as shown below.

[Figure]

**Fig. 2** Diurnal variations in concentrations of some reactive VOCs species in Zhengzhou under different pollution levels.The light blue, light red, and light green shadows represent stages P1, P2, and P3, respectively.

[Figure]

**Fig. S5** Diurnal variation distribution of pollutants during different pollution periods. The light blue, light red, and light green shadows represent stages P1, P2, and P3, respectively.

11. Lines 409 – 412: How did the author determine the correlation between age indicator and O3 as strong?

**Response:** Thank you for your feedback. Through linear regression analysis of ozone and E/X, a significant correlation was observed under different levels of ozone pollution, with R² values greater than 0.7, confirming their strong relationship (as shown in Figure S5).

12. Lines 621: Why was the HONO mixing ratio underestimated? Can it be overestimated? Same question for the HOX.

**Response:** Thank you for your feedback. Since we did not test HONO, the use of the OBM model for simulations leads to an underestimation of the radical concentrations.

13. Figure 9: Could the authors label NOx and VOC control zones in the plot? Where is the transition regime?

**Response:** As shown in Figure 9,the ridge line, formed by the turning points of the iso-lines, divides the curve into two distinct regions. Above the ridge line lies the VOCs-controlled region, where ozone concentrations are more sensitive to VOC variations, with little influence from NOx. Below the ridge line is the NOx-controlled region, where ozone levels are primarily influenced by NOx changes, with minimal impact from VOCs. Near the ridge line, the transition regime exists, where both VOCs and NOx need to be adjusted to effectively reduce ozone levels.

**Technical Comments:**

1. Make consistent formatting for "NOx" and "HOx".

**Response:** Thank you for pointing this out. I have ensured consistent formatting for both "NOx" and "HOx" throughout the manuscript.

2. Lines 57 – 58: Provide a few references to support the sentence

**Response:** Thank you for your suggestion. The introduction has been rewritten.

3. Line 97: Use another word to replace "escalating".

**Response:** Thank you for your suggestion. The introduction has been rewritten.

4. Line 136: The abbreviation "VOC" can be used here.

**Response:** Thank you for your suggestion. The introduction has been rewritten.

5. Line 201: Should j be in italic?

**Response:** I have italicized "j" in line 201 as per your suggestion, in accordance with standard scientific notation

6. Lines 213 – 224: They symbols should be consistent in both paragraphs and equations.

**Response:** Accepted.

7. Line 223: Should k be in italic?

**Response:** I have italicized "k" as per your suggestion, in accordance with standard scientific notation

8. Figure S1: What does the small red star stand for? Also, the star is too small to read.

**Response:** Thank you for your suggestion. The small red stars in the figure indicate the locations of the observation stations. Figure S1 has been updated.

[Figure]

**Fig. S1** Locations of the sampling stations in Zhengzhou.

9. Figure S2: What are d01, d02, d03 and d04? I could not find them in the figure.

**Response:** I sincerely apologize for any inconvenience caused. Figure S2 has been updated. D1, D2, D3 and D4 have horizontal resolutions of 36, 12, 4 and 1 km, respectively.

[Figure]

**Fig. S2** Four-level nested domains used in the WRF/CMAQ simulations. D1, D2, D3 and D4 have horizontal resolutions of 36, 12, 4 and 1 km, respectively.

10. Line 264: more accurate than what?

**Response:** Thank you for your valuable feedback. This section has been rewritten as follows: "On the contrary, the simulation of nitrogen dioxide is generally more accurate, as $NO_2$ is directly emitted, leading to better model performance in simulating $NO_2$."

11. Line 291: What does "quality of species" mean?

**Response:** I apologize for the mistake. This has been corrected to concentration.

12. Line 314: Is it supposed to be "ErrorFraction" instead of "Error"?

**Response:** Thank you for your suggestion; it has been corrected.

13. Line 372: similar pattern to what?

**Response:** Thank you for your suggestion. This sentence has been updated to: "$PM_{10}$ exhibited a pattern similar to that of NOx."

14. Figure 2 and Figure S5: Please use the interquartile range as the error bars.

**Response:**Thank you very much for your valuable feedback. Regarding the error bars, the figures (Figure 2 and Figure S5) present the daily variation distributions of pollutant concentrations under different levels of pollution. Adding error bars would make the figures more cluttered and may hinder the clarity of the information being conveyed. To maintain the simplicity of the figure and to ensure a clear comparison between the different pollution levels, we have chosen not to include error bars at this stage. We believe that presenting all the data in a single figure allows for a more intuitive comparison of the different pollution levels.

However, if you believe that including error bars would enhance the accuracy and readability of the figures, we would be happy to consider splitting the figure into separate ones, adding error bars to each, while still aiming to preserve the comparative effect.Once again, thank you for your feedback, and we look forward to your further guidance.

15. Line 416: What concentration decreased to a minimum?

**Response:**Thank you for your valuable feedback. It should be the $O_3$ concentration here.

16. Line 453: What is a coordinated control zone?

**Response:**Thank you for your suggestion. To avoid any confusion, this has been corrected to "transition zones."

17. Line 497: I am confused with the part "in order to mitigate O3 pollution under unfavorable conditions."

**Response:**Thank you for your valuable feedback. The sentence has been deleted to avoid any ambiguity.

18. Figure 4: What do those dots and bars stand for?

**Response:**In Figure 4, the dots represent the proportion of individual substances in the concentration of each factor, while the bars represent the concentration of the substance.

19. Line 595: It should be "cm3"

**Response:**Thank you for your suggestion; the changes have been made.

20. Table S2: Where is the data for highly polluted days?

**Response:**Due to the limited number of heavy pollution days (less than 5), there are fewer samples. Therefore, moderate and heavy pollution days were combined for statistical analysis.

21. Line 638: Is it "moderately/highly polluted days" or "moderately and highly polluted days"?

**Response:**Due to the limited number of heavy pollution days (less than 5), there are fewer samples. Therefore, moderate and heavy pollution days were combined for statistical analysis. Therefore, it should be "moderately/highly" here.

---

## Author Comment (AC2)

**Manuscript number:** EGUSPHERE-2024-4178

**Title:** Diagnosing O₃ formation and O₃-NOX-VOC sensitivity in a heavily polluted megacity of central China: A multi-method systematic evaluation over the warm seasons from 2019 to 2021

This manuscript presents a comprehensive investigation of O3 pollution in Zhengzhou over the warm seasons from 2019 to 2021, utilizing observational data and model simulations (CMAQ, PMF, and OBM) to provide insights into the O3-NOx-VOC sensitivity and to propose effective control strategies. Significant improvements are needed in terms of novelty, logistics, and writing. Below are my main comments.

> **Response:** We appreciated it very much for the reviewer's positive comments and valuable suggestions. Below are the point-to-point responses to all of the comments (The comments are marked in black color and the responses are marked in dark blue color). The major changes that have been made according to these responses were marked in yellow color in the highlighted copy of the revised manuscript. And our own minor changes were marked in red font. Note that the following line numbers are shown in the corrected version.

**Major comments**

1. More comprehensive analysis of O3 pollution characteristics in Zhengzhou. The title describes Zhengzhou as "a heavily polluted megacity of central China," yet the manuscript does not provide sufficient evidence to justify this statement. Additionally, comparisons with other urban areas should be incorporated into the results discussion.

**Response:** Thanks for your suggestions. Zhengzhou is a typical heavily polluted city in central China, facing severe air pollution issues in recent years. According to various studies, the air quality in Zhengzhou has consistently been poor, especially with high concentrations of pollutants such as $PM_{2.5}$ and $O_3$. The introduction of this paper will briefly outline the pollution status of Zhengzhou, further confirming its classification as a heavily polluted city through the reference of relevant literature. In section 2.1 of the methods, we have focused on explaining Zhengzhou as a major polluted megacity in central China.

In chapters concerning the concentration characteristics of VOCs and VOCs source analysis, we have compared and analyzed domestic and international research findings to enrich the manuscript. By incorporating studies from different countries and regions, a more comprehensive understanding of Zhengzhou's pollution

characteristics and trends can be achieved, thus providing more accurate evidence for pollution control. The updates and tables are shown below.

Table S3 presents a comparison between the concentration characteristics of VOCs in this study and those reported in domestic and international literature. The concentration of VOCs in Zhengzhou (90.3 ± 52.8 μg/m³; 2019-2021) is higher than that of most international urban regions, aligning with other regions in China, such as Beijing (101.5 ± 65.2 μg/m³; 2016) (Liu et al., 2021). In contrast, cities like Istanbul (40-60 μg/m³), Athens (50.3 μg/m³; 2016-2017), and Vitória, Brazil (24.1 ± 29.6 μg/m³; 2022-2024) show significantly lower levels in the concentration of VOCs, reflecting regional disparities in emission control and industrial development (Thera et al., 2019; Panopoulou et al., 2021; Galvão et al., 2025). The VOCs in Zhengzhou are mainly composed of dichloromethane, acetone, ethane, isopentane, and n-hexane, indicating mixed sources from solvents and vehicular emissions, different from other cities where industrial and traffic emissions are more specialized. This finding suggests that there is an urgent need for targeted abatement efforts to reduce the concentration of VOCs and alleviate ozone pollution in a rapidly growing city like Zhengzhou.

**Table S3** Comparative analysis of VOCs pollution characteristics in Zhengzhou and other cities

| No. | Time Period | Location | Site Type | VOCs Concentration | Dominant VOCs Species / Composition | Reference |
|---|---|---|---|---|---|---|
| 1 | May-Sep 2019-2021 | Zhengzhou City, China | Urban site | 90.3 ± 52.8 μg/m³ | Dichloromethane, Acetone, Ethane, Isopentane, n-Hexane | This study |
| 2 | Summer 2019 | Coastal Tokyo Bay, Japan (Yokohama area) | Industrial site (Ushioda) | 86 μg/m³ (Range: 29–199 μg/m³) | Propane (6.9 μg/m³) > Ethyl acetate (6.0 μg/m³) > n-Butane (5.2 μg/m³) > Ethane (5.0 μg/m³) > Ethyl acetate (4.8 μg/m³) | Fukusaki et al., 2021 |
| | | Port-industrial mixed area (Yokohama Tower) | Port-industrial mixed site | 70 μg/m³ (Range: 20–255 μg/m³) | Toluene (5.1 μg/m³) > Propane (4.9 μg/m³) > Ethyl acetate (4.8 μg/m³) > Ethyl acetate (3.8 μg/m³) > Ethane (3.5 μg/m³) | |
| 3 | Jan-Nov 2010 | Greater Paris Metropolitan Area, France | Urban site | / | Alkanes: 39.1%, Oxygenated VOCs: 36.5%, Aromatics: 16.9%, Alkenes + Alkynes + Dienes: 7.5% | Baudic et al., 2016 |
| 4 | [Data missing] | Beşiktaş District, Istanbul, Turkey | Urban site | 40–60 μg/m³ (Average range) | Oxygenated VOCs: 43.9% (≈17 μg/m³), Alkanes: 26.3% (≈16 μg/m³), Aromatics: 20.7% (≈18 μg/m³), Alkenes: 4.8% (≈3 μg/m³) | Thera et al., 2019 |
| 5 | Mar 2016 – Feb 2017 | Athens, Greece | Urban site | 50.3 μg/m³ | Toluene: 6.7 μg/m³, Isopentane: 6.5 μg/m³, Ethane: 4.2 μg/m³, m/p-Xylene: 4.0 μg/m³, n-Butane: 3.8 μg/m³ | Panopoulou et al., 2021 |
| 6 | Jul 2022 – Apr 2023; Dec 2023 – Mar 2024 | Vitória, Brazil | Urban site | 24.1 ± 29.6 μg/m³ | n-Pentane (4.59 μg/m³), Benzene (3.09 μg/m³), 1,2,4-Trimethylbenzene (1.49 μg/m³), Ethylbenzene (2.41 μg/m³) | Galvão et al., 2025 |

| 7 | 2017 | Taiwan | Urban & industrial sites | 53.4–76.0 μg/m³ | Alkanes (46.5–55.3%) > Aromatics (28.0–42.2%) > Alkenes (7.4–11.4%) > Alkynes (1.2–5.5%) | Huang et al., 2020 |
| 8 | Jan-Jul 2016 | Beijing, Jing-Jin-Ji Region, China | Urban site | 101.5 ± 65.2 μg/m³ | Alkanes > OVOCs > Halogenated hydrocarbons > Aromatics > Alkenes > Alkynes | Liu et al., 2021 |
| 9 | Jul-Dec 2019 | Hefei, Yangtze River Delta Region, China | Urban site | 68.79 μg/m³ | Alkanes > OVOCs > Halogenated hydrocarbons > Aromatics > Alkenes > Alkynes | Wang et al., 2022 |

Table S4 presents a comparison of source apportionment between Zhengzhou and other cities. During the observation period, the main sources of VOCs in Zhengzhou comprise vehicle emissions (31%), solvent use (24%), and industrial processes (21%), collectively accounting for 76% of the total pollution, highlighting the dominant role of traffic and industrial pollution. In contrast, cities like Paris and Turkey have significantly lower proportions of vehicle emissions (15% and 15.8%, respectively) (Baudic et al., 2016; Thera et al., 2019). The proportion of solvent use in Zhengzhou (21%) is similar to that in the cities of the Yangtze River Delta but higher than that in Turkey and other regions (Zhang et al., 2025; Thera et al., 2019). The severity of biogenic pollution in Zhengzhou is lower, a phenomenon particularly evident in cities with richer vegetation cover, such as Athens (Kaltsonoudis et al., 2016). The source apportionment structure in Zhengzhou reflects its typical characteristics as an industrial city, with significant pressure from traffic emissions, as well as notable contributions from solvent use and industrial processes, while biogenic sources contribute relatively little.

**Table S4** Source analysis of VOCs in Zhengzhou and comparison with other cities.

| No. | City/Location | Study Period | Site Type | PMF Source Apportionment (%) | Reference |
|---|---|---|---|---|---|
| 1 | Central Plains (Zhengzhou, Henan), China | Jan 2019-Sep 2021 | Urban | Vehicle emissions: 32.4%; Solvent use: 24.8%; Industrial processes: 18.3%; LPG/NG combustion: 12.6%; Combustion sources: 8.9%; Biogenic: 3.0% | This study |
| 2 | Jing-Jin-Ji Region (Beijing), China | 1 Jun-31 Aug 2020 | Urban | Gasoline exhaust: 23.5%; Biogenic: 19.2%; Fuel evaporation: 15.7%; Diesel exhaust: 15.2%; Solvents: 14.3%; Industrial processes: 12.1% | Li et al., 2024 |

| | | | | | |
|---|---|---|---|---|---|
| 3 | North China (Changzhi, Shanxi), China | 2-30 Jun 2021 | Urban | Gasoline vehicles: 27.0%; Coal combustion: 20.3%; Diesel vehicles: 15.9%; Industrial processes: 15.1%; Solvents: 14.0%; Biogenic: 7.6% | Niu et al., 2024 |
| 4 | Yangtze River Delta (Suzhou, Jiangsu), China | 2015-2022 | Urban | Fossil fuel combustion: 20.4%; Solvents: 17.7%; Gasoline evaporation: 16.7%; Diesel exhaust: 12.6%; Natural gas: 10.8%; Industrial: 9.7%; Vehicle exhaust: 7.7%; Biogenic: 4.5% | Zhang et al., 2025 |
| 5 | Yangtze River Delta (Tongxiang, Zhejiang), China | 1 May-25 Jul 2021 | Urban | Solvents & gasoline: 32.3%; Temperature-dependent sources: 28.1%; Vehicle exhaust: 19.9%; Manufacturing: 14.4%; Petrochemical: 7.8% | Qu et al., 2025 |
| 6 | Taiwan Region (Taipei), China | Mar 2020-Feb 2021 | Urban | Industrial solvents: 11.97-37.62%; Household emissions: 14.67-31.62%; Biogenic: 1.7-25.21%; Diesel vehicles: 5.74-19.03%; Petrochemicals: 5.43-18.67%; Gasoline vehicles: 4.51-12.34% | Chen et al., 2023 |
| 7 | Vitória, Brazil (ES Site) | Jul 2022-Apr 2023; Dec 2023-Mar 2024 | Urban site | Vehicle exhaust: 46%; Coke ovens: 26%; Solvents: 13%; Industrial processes: 11%; Fuel evaporation: 4% | Galvão et al., 2025 |
| 8 | Paris Metropolitan Area, France | 15 Jan-22 Nov 2010 | Urban | Natural gas + background: 23%; Solvents: 20%; Wood burning: 17%; Vehicle exhaust: 15%; Biogenic: 15%; Gasoline evaporation: 10% | Baudic et al., 2016 |
| 9 | Athens, Greece | 3-26 Jul 2012 (Summer) | Urban | Traffic: 37.1%; Biogenic VOCs: 26.2%; Secondary oxidized VOCs: 19.3%; Biogenic oxidized VOCs: 18.4% | Kaltsonoudis et al., 2016 |
| 10 | Beşiktaş District, Istanbul, Turkey | 14-30 Sep 2014 | Urban | Mixed area emissions: 36.3%; Natural gas evaporation: 25.9%; Road transport: 15.8%; Solvents (toluene): 14.2%; Biogenic terpenes: 7.8% | Thera et al., 2019 |

2. More details on the CMAQ model configuration. The current version lacks a detailed description of the CMAQ model configuration. Essential aspects such as horizontal and vertical resolution, meteorological condition, chemical mechanisms, emission inventories, and boundary conditions should be explicitly stated. Additionally, Fig.S2 is too blurred, and the information it expresses should be described in detail in manuscript.

**Response:** Thank you very much for your valuable feedback. I have added more details about the CMAQ model configuration in the manuscript and summarized them in Table S1. Additionally, Figure S2 has been updated. The revised manuscript and figures are presented below.

"The decoupled direct method (DDM) is simulated using the WRF/CMAQ model, with detailed configuration information provided in the papers published by our research group. Additionally, the WRF/CMAQ setup is summarized in Table S1. More specifically, the WRF/CMAQ model is configured with four nested domains: 36 km

for East Asia, 12 km for central and eastern China, 4 km for Henan Province, and 1 km for Zhengzhou (Fig. S2). WRF provides meteorological inputs for CMAQ, using 6-h FNL global reanalysis data as initial and boundary conditions. The SAPRC-99 gas-phase photochemical mechanism and AERO6 aerosol module are utilized in the CMAQ model, with modified heterogeneous chemistry for SO2 to sulfate and NO2 to nitrate conversion (Hu et al., 2014). The clean continental IC/BC is used in the 36-km simulation, and the nested domain IC/BC is derived from the parent domains. The CMAQ output within the first five days is discarded to minimize IC influence. The anthropogenic emission data of China are obtained based on the 2016 MEIC inventory (at a resolution of 0.25° × 0.25°), while the emission data of other regions are collected from the REAS2 inventory. The emission data of Henan Province (4-km domain) are acquired based on the local data from Bai et al. (2020). The biogenic emission data of all domains are generated using MEGAN (version 2.10), with the windblown dust emission data generated online in CMAQ simulations."

[Figure]

**Fig. S2** Four-level nested domains used in the WRF/CMAQ simulations. D1, D2, D3 and D4 have horizontal resolutions of 36, 12, 4 and 1 km, respectively.

**Table S1** Key parameter settings for the mode section.

| Main parameters | Content |
|---|---|
| WRF Vertical resolution (eta_levels) | 1.000, 0.996, 0.99, 0.98, 0.97, 0.96, 0.95, 0.94, 0.93, 0.92, 0.91, 0.895, 0.88, 0.865, 0.85, 0.825, 0.8, 0.775, 0.75, 0.72, 0.69, 0.66, 0.63, 0.6, 0.57, 0.54, 0.51, 0.475, 0.44, 0.405, 0.37, 0.33, 0.29, 0.25, 0.21, 0.175, 0.145, 0.115, 0.09, 0.065, 0.045, 0.025, 0.01, 0.000 |
| Microphysics scheme | New Thompson et al. (scheme 8) |

| Longwave radiation | Rapid Radiative Transfer Model |
| Shortwave radiation | Goddard shortwave scheme |
| Land-surface scheme | Unified Noah land-surface model |
| Surface layer scheme | MYJ surface scheme |
| Boundary layer parameterization | Yonsei University scheme (Non-local -K scheme 1) |

3. Different methods were included, including CMAQ, PMF and OBM. However, no clear connections and intercomparison were introduced for these methods. Were these methods really necessary?

**Response:** Thank you for your careful review and valuable comments. Regarding your suggestion about the connection between model selection and methodology, I have carefully considered it and made corresponding adjustments and additions. Below is my response:

This study is based on three years of real-time observational data from Zhengzhou and utilizes a combination of multiple models (CMAQ, PMF, and OBM) to identify the characteristics of ozone precursor pollutants and their influence on ozone formation. We selected these models based on their respective strengths and aligned them with the specific research objectives. Specifically, CMAQ and PMF are used for source apportionment of ozone and VOCs, respectively. Although PMF is widely used in source apportionment, it has been insufficiently applied in ozone source apportionment studies. Therefore, this paper strengthens the research on the source apportionment of ozone and its precursors, and the findings show that ozone and its precursors in Zhengzhou are mainly influenced by motor vehicle emissions, which is one of the innovative contributions of this study.

Furthermore, in the study of ozone sensitivity, we employed a variety of methods, including the ratio method, OBM's RIR and EKMA methods, and the CMAQ's DDM method. By coupling these multiple approaches, we aim to overcome the limitations of any single method. We recognize that each method has its advantages and limitations, and thus combining them allows for a more comprehensive understanding of the ozone sensitivity characteristics.

4. Describe the main improvement or innovation compared with your previous study.

Wang, X. D., Yin, S. S., Zhang, R. Q., Yuan, M. H., and Ying, Q.: Assessment of summertime O3 formation and the O3-NOx-VOC sensitivity in Zhengzhou, China using an observation-based model, Sci. Total Environ., 813, 152449, 2022.

**Response:** This study introduces several improvements and innovations compared to the previous research conducted by members of our research group:

The previous study relied on data collected over a one-month period, capturing three instances of ozone pollution characteristics. However, the short duration of this dataset introduced a higher degree of randomness and larger potential errors. In contrast, this study utilizes three years of observation data and, following the GB3095-2012 standard, systematically classifies ozone pollution levels. We comprehensively analyzed the characteristics and formation mechanisms of ozone and its precursors under different pollution levels, ensuring more reliable and robust results.

This paper goes beyond the traditional approach of source apportionment for volatile organic compounds (VOCs) by conducting a detailed source tracing for both ozone and its precursors. Unlike previous studies that focused solely on VOCs, our research offers a more holistic understanding of the sources of ozone in Zhengzhou, providing critical data for targeted emission reduction strategies.

While the previous study used the OBM model to assess ozone sensitivity, this research takes a more comprehensive approach by incorporating the ratio method, OBM model, and CMAQ-DDM methods. This multi-method approach enables a more thorough and integrated evaluation of ozone sensitivity, providing a broader perspective on the findings.

5. In referencing previous studies, should write as Huang et al. (2019) instead of Huang (2019) (line 73). Do this for the other references.

**Response:** Thank you for your detailed review and valuable feedback. Regarding the reference format issue you raised, we have made the necessary revisions based on your suggestion. Specifically, we have changed "Huang (2019)" to "Huang et al. (2019)" throughout the manuscript, including in line 73, and have made similar adjustments to the other references.

**Detailed comments:**

1. Line 40-41: The phrase "precursor emissions" is inaccurate, please revise for clarity.

**Response:** Thank you for your valuable comments. Regarding your feedback on the phrase "precursor emissions," we agree that this term may not be clear in the context. To improve clarity, we have revised the sentence accordingly. Specifically, we have

removed "precursor emissions" and clarified the message by directly referring to "VOC emissions." Revised sentence:

"The results demonstrated that reducing vehicle emissions should be prioritized to mitigate ozone pollution in Zhengzhou, as transportation emissions accounted for 64% and 31% of ozone and VOC emissions, respectively."

2. Line 42: Should be "observation-based model (OBM)".

**Response:** We have revised the phrase to "observation-based model (OBM)" as recommended.

3. Line 59: "Continue to increase" should be revised to "continue increasing."

**Response:** Accepted.

4. Line 61: Clarify the distinction between "VOC" and "VOCs" and explain why this differentiation is necessary.

**Response:** Sorry for the misunderstanding. "VOCs" is typically used to refer to a group of volatile organic compounds, as it represents a collection of different chemical substances. We have made the necessary revisions throughout the manuscript to ensure consistency, and all instances have been updated to "VOCs."

5. Lines 71–72: The sentence structure is overly complex.Lines 83: can provide abbreviation for "Yangtze River Delta" here and then use it later.

**Response:** Thank you for your suggestion. The introduction has been rewritten, and I have hired a professional editing company to polish it.

6. Lines 264-265: I do not understand the logic here. It's not surprising that the model has a better performing in simulating NO2 since it was directly emitted. The logic does not make sense at all here.

**Response:** Apologies for any confusion caused. Based on your suggestion, this sentence has been corrected to: On the contrary, the simulation of nitrogen dioxide is generally more accurate, as $NO_2$ is directly emitted, leading to better model performance in simulating $NO_2$."

7. Line 325: "specie" should be corrected to "species".

**Response:** Thank you for your suggestion; the correction has been implemented as advised.

8. Line 338: Please define MDA8 correctly.

**Response:** Thank you for your suggestion. Following the published literature, this section has been updated to "the maximum daily average 8-hour (MDA8) O3 concentrations."

9. Line 384: Clearly define "non-polluted," "lightly polluted," and "moderately polluted" periods.

**Response:** Thank you very much for your suggestion. The definitions for different pollution scenarios have been established, and this section has been updated to:

"According to the GB 3095-2012 standard, MDA8 O3 concentrations exceeding 160 and 215 μg/m³ are defined as light pollution and moderate pollution, respectively. The average VOCs concentrations for non-pollution, light pollution, and moderate pollution periods were 84.7±51.0, 96.6±53.4, and 105.3±59.4 μg/m³, respectively."

10. Line 402: Mark the P1–P4 phases in Fig. S5 to improve readability.

**Response:** Thank you for your suggestions. The Fig. S5 has been redrawn as shown below.

[Figure]

**Fig. S5** Diurnal variation distribution of pollutants during different pollution periods. The light blue, light red, and light green shadows represent stages P1, P2, and P3, respectively.

11. Line 474: How did the authors conclude that "cross-regional mitigation measures" are required? Please provide a clear rationale.

**Response:**Thank you for your valuable comment. We concluded that "cross-regional mitigation measures" are required based on the observation of relatively high O₃ concentrations in areas surrounding Zhengzhou, as shown in Fig. 3b. These high concentrations in neighboring regions suggest that ozone pollution is not confined to a single locality but is instead a regional issue, likely influenced by factors such as wind patterns, transportation, industrial emissions, and urbanization, which affect multiple areas within Henan Province.

O₃ pollution in Zhengzhou and surrounding areas is interconnected, meaning that measures taken in one region alone may not be sufficient to effectively reduce the overall O₃ levels. Therefore, to achieve meaningful reductions in ozone concentrations, coordinated efforts across regions are necessary, addressing both local emissions and regional transport of pollutants. This would require collaboration between neighboring regions to implement effective air quality management strategies, such as controlling emissions from transportation and industries that affect multiple areas simultaneously.

12. Line 496: Clarify whether the "power sector" is equivalent to "electricity".

**Response:**Thank you for your valuable comment. We confirm that "power sector" is equivalent to "electricity" in this context. To avoid any potential ambiguity, we have standardized the term to "electricity" and made the corresponding revisions in the text.

13. Lines 713–714: Please label the relevant regions in Fig. 10 to enhance readability.

**Response:**Thank you very much for your suggestions. I have updated Fig. 10 based on your feedback.

[Figure]

**Fig. 10** Spatial comparison of O₃-NOx-VOCs sensitive regime from 2019 to 2021 in

Zhengzhou.

14. Conduct a thorough grammatical revision and refine sentence structures throughout the manuscript.

**Response:** Thank you for your feedback. The manuscript has been professionally edited by a specialized agency, and here is the proof of editing.

[Figure]

**CERTIFICATE OF EDITING**

This is to certify that the paper titled **Diagnosing O₃ formation and O₃-NOX-VOC sensitivity in a heavily polluted megacity of central China: A multi-method systematic evaluation over the warm seasons from 2019 to 2021** commissioned to us by **Shijie Yu, Hongyu Liu, Hui Wang, Fangcheng Su, Beibei Wang, Minghao Yuan, Kunao Song, Zixian Wang, Daoqing Xu, Ruiqin Zhang** has been edited for English language, grammar, punctuation, and spelling by Enago, the editing brand of Crimson Interactive Consulting Co., Ltd..

✓ **ISO 17100:2015**
Translation Service Providers

✓ **ISO 27001:2013**
Information Security Management System

✓ **ISO 9001:2015**
Quality Management System

Issued by:
Enago, Crimson Interactive (Beijing) Consulting Co., Ltd.
Room 3217, Cyber Tower A, No. 2,
Zhongguancun South Street,
Haidian District, Beijing

Disclaimer: The intent of the author's message has been preserved during the editing process. The author is free to accept or reject our changes in the document after reviewing our editing. This certificate has been awarded at the time of sharing the final edited version (full file or sections of the file) with the author. Enago does not bear any responsibility for any alterations done by the author to the edited document post 23ʳᵈ Nov. 2024.

| | | |
|---|---|---|
| Japan | www.enago.jp, www.ulatus.jp, www.voxtab.jp | |
| China | www.enago.cn, www.ulatus.cn | Russia www.enago.ru |
| Brazil | www.enago.com.br, www.ulatus.com.br | Arabic www.enago.ae |
| Germany | www.enago.de | Turkey www.enago.com.tr |
| | | S.Korea www.enago.co.kr |
| | | Global www.enago.com, www.ulatus.com, www.voxtab.com |

About Crimson:
Crimson Interactive Consulting Co. Ltd. is one of the world's leading academic research support services. Since 2005, we've supported over 2 million researchers in 125 countries with their publication goals.